# Enhanced light absorption for solid-state brown carbon from wildfires due to organic and water coatings

Zezhen Cheng [1], Manish Shrivastava [2], Amna Ijaz[1,2,3], Daniel Veghte[1,4], Gregory W. Vandergrift[1], Kuo-Pin Tseng[1,5], Nurun Nahar Lata[1], Will Kew [1], Kaitlyn Suski[2,6], Johannes Weis[7], Gourihar Kulkarni [2], Larry K. Berg [2], Jerome D. Fast [2], Libor Kovarik[1], Lynn R. Mazzoleni[3], Alla Zelenyuk [2] & Swarup China [1] ✉

Wildfires emit solid-state strongly absorptive brown carbon (solid S-BrC, commonly known as tar ball), critical to Earth's radiation budget and climate, but their highly variable light absorption properties are typically not accounted for in climate models. Here, we show that from a Pacific Northwest wildfire, over 90% of particles are solid S-BrC with a mean refractive index of 1.49 + 0.056$i$ at 550 nm. Model sensitivity studies show refractive index variation can cause a ~200% difference in regional absorption aerosol optical depth. We show that ~50% of solid S-BrC particles from this sample uptake water above 97% relative humidity. We hypothesize these results from a hygroscopic organic coating, potentially facilitating solid S-BrC as nuclei for cloud droplets. This water uptake doubles absorption at 550 nm and the organic coating on solid S-BrC can lead to even higher absorption enhancements than water. Incorporating solid S-BrC and water interactions should improve Earth's radiation budget predictions.

Light-absorbing carbon, including black carbon (BC) and brown carbon (BrC), in wildfire smoke has critical climate warming effects[1,2]. BC is typically considered the dominant light absorber[2], but studies have reported that the wildfire smoke particles contain a special type of solid-state strongly absorptive BrC (solid S-BrC) commonly known as tar balls[3–5]. The number fraction of solid S-BrC in the wildfire smoke particles can vary from <10% to >95% depending on the transport distance and atmospheric aging[6–9]. Moreover, the imaginary part of the refractive index (RI) at 550 nm wavelengths ($k_{550}$) reported for solid S-BrC varies over two orders of magnitude (between $10^{-3}$ and $10^{-1}$), which might be due to the bias in various analytical techniques as well as complex chemical nature and evolution of chemical properties during atmospheric aging and transport[6,7,10–18]. Besides, solid S-BrC is

commonly emitted with other particles, such as soot, other organic aerosol (other OA) which are not solid S-BrC, and inorganic particles[1,19,20], which might misrepresent solid S-BrC optical properties based on bulk optical measurements. Furthermore, little is known about the interaction between solid S-BrC and water[15,21–23]. For example, Semeniuk et al.[21] and Adachi and Buseck[22] show negligible hygroscopic growth of solid S-BrC at 100% relative humidity (RH). However, Hand et al.[15] reported that solid S-BrC can uptake water at ~83% RH. Adachi et al.[23] reported abundant solid S-BrC collected in the pyrocumulonimbus cloud to have thin layers or coatings as residual of water mixed with water-soluble species, suggesting solid S-BrC could be hygroscopic and potential nuclei for cloud droplets. These studies show discrepancies in solid S-BrC hygroscopicity, and the relative

[1]Environmental Molecular Sciences Laboratory, Pacific Northwest National Laboratory, Richland, WA, USA. [2]Atmospheric, Climate, & Earth Sciences Division, Pacific Northwest National Laboratory, Richland, WA, USA. [3]Department of Chemistry, Michigan Technological University, Houghton, MI, USA. [4]The Ohio State University, Columbus, OH, USA. [5]University of Illinois at Urbana-Champaign, Champaign, IL, USA. [6]Rainmaker Technology Corporation, El Segundo, CA, USA. [7]Chemical Sciences Division, Lawrence Berkeley National Laboratory, Berkeley, CA, USA. ✉e-mail: swarup.china@pnnl.gov

abundance of hygroscopic and hydrophilic solid S-BrC in the atmosphere is still missing. Moreover, it has been reported that coating on soot can enhance the light absorption properties of soot[24,25]. However, the effects of these coatings on solid S-BrC light absorption properties have not been investigated, which might contribute to the aerosol optical properties discrepancy between models and observations. Therefore, the regional climate effects of wildfire smoke have significant uncertainties due to unresolved variability of solid S-BrC concentration, optical properties, and hygroscopicity.

Here, we report a comprehensive single-particle and molecular-level analysis of solid S-BrC particles collected during the Pacific Northwest wildfire events on September 5 and 6, 2017, where >90% of particles were solid S-BrC. Given this composition, this event provides a unique opportunity to probe the physical, chemical, and optical properties of solid S-BrC. The experimentally retrieved solid S-BrC optical properties and mass fractions were used as inputs to the Weather Research and Forecasting Model coupled to chemistry (WRF-Chem) to estimate their absorption aerosol optical depth (AAOD) over the Pacific Northwest region. Additionally, we investigated the interactions between solid S-BrC and estimated the lensing enhancement due to water coating. Our results show that solid S-BrC can dominate in wildfire smoke. We found that ~50% of solid S-BrC particles can uptake water above 97% RH, which results in a lensing enhancement at 550 nm by a factor of 2. Furthermore, the light-absorbing organic can coat solid S-BrC, leading to even higher absorption enhancements than

water. Additionally, we compare compositional results from a wildfire-impacted plume in the Pacific Northwest (August 2018) to assess the broader applicability of our findings across the region[26].

## Results

### Prevalence of solid S-BrC in wildfire smoke aerosol

An accurate estimate of the solid S-BrC number fraction in wildfire smoke is critical for predicting its climate effects. Figure 1a shows the representative 75° tilted scanning electron microscope (SEM) image of the wildfire particles sample. We manually identified and counted the number fraction of solid S-BrC (spherical shape) and other OA particles that are not solid S-BrC (other OA, dome-like or flat shape), BC particles (fractal or compressed small monomer aggregates), and inorganic particles (crystal or irregular shape)[20,27]. More than 90% of particles in our samples were solid S-BrC (Fig. 1a). Moreover, 34% of solid S-BrC in these samples are fractal-like aggregates, which might be formed by aggregation during atmospheric aging and transport[28].

Figure 1b shows the chemically resolved size distribution of over 3000 particles derived from computer-controlled scanning electron microscopy with an energy-dispersive X-ray spectrometer (CCSEM-EDX). The carbonaceous (CNO), sulfur-containing carbonaceous (CNOS), and potassium-containing carbonaceous (CNOK) particles are dominant in the sample (~92%). Because tilted SEM images show a negligible number of inorganics (<1%) and BC (<1%) and dominated solid S-BrC (>90%), the CNO, CNOS, and CNOK particles are likely solid

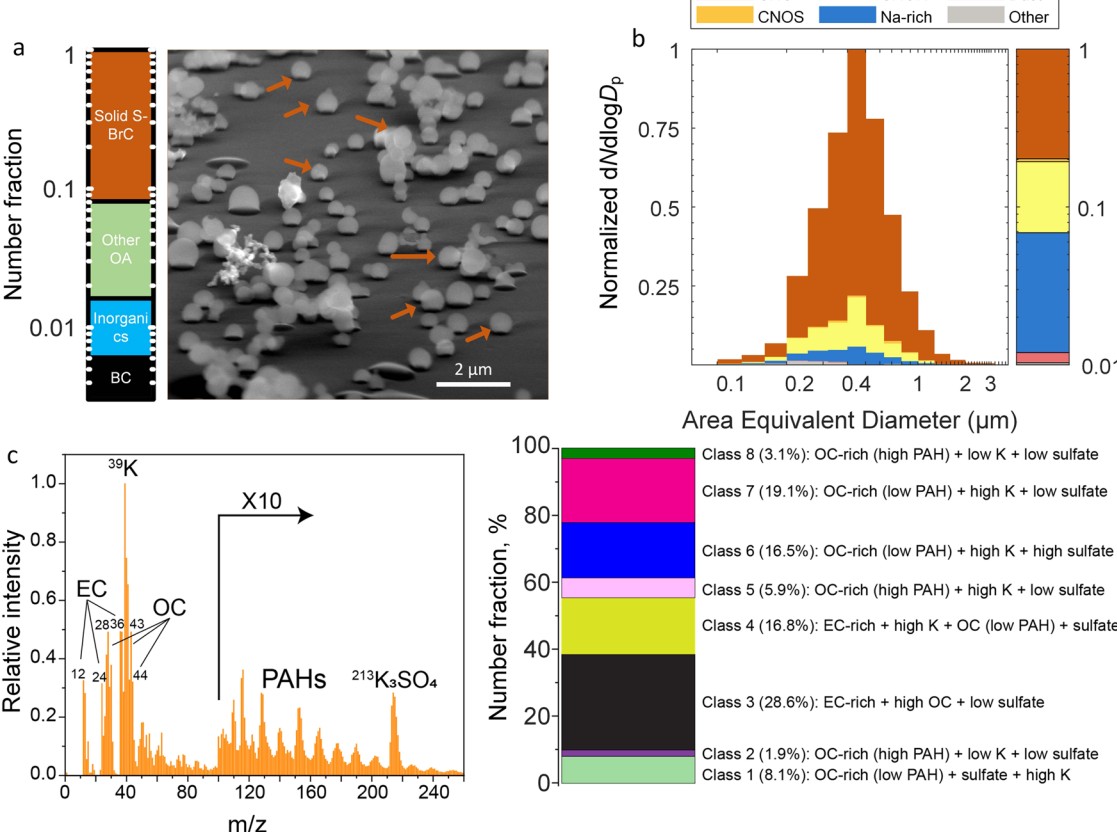

**Fig. 1 | Comprehensive characterization of individual particles from microscopy and single particle mass spectrometry. a** Representative 75° tilted SEM image of particles. Dark brown arrows indicate some solid-state strongly absorptive BrC (solid S-BrC) as examples. The bar chart shows the fraction of different types of particles based on manual SEM identification (solid S-BrC (spherical): ~93%, other organic aerosol (OA, dome-like or flat shapes): ~6%, black carbon (BC, fractal or compressed small monomer aggregates): <1%, and inorganics (crystal or irregular

shapes): <1%). **b** CCSEM-EDX-derived chemically resolved size distribution of wildfire smoke aerosol. Size distribution indicates carbonaceous (CNO) particles dominate with a mode diameter of ~0.4 μm. The presence of potassium-containing carbonaceous (CNOK) particles is an indicator of wildfire. The size mode at 400 nm is due to the solid S-BrC aggregates. **c** The average mass spectrum of all particles sampled and characterized by miniSPLAT. The bar plot is the number fraction of 8 solid S-BrC classes based on the miniSPLAT mass spectra (see Section S1).

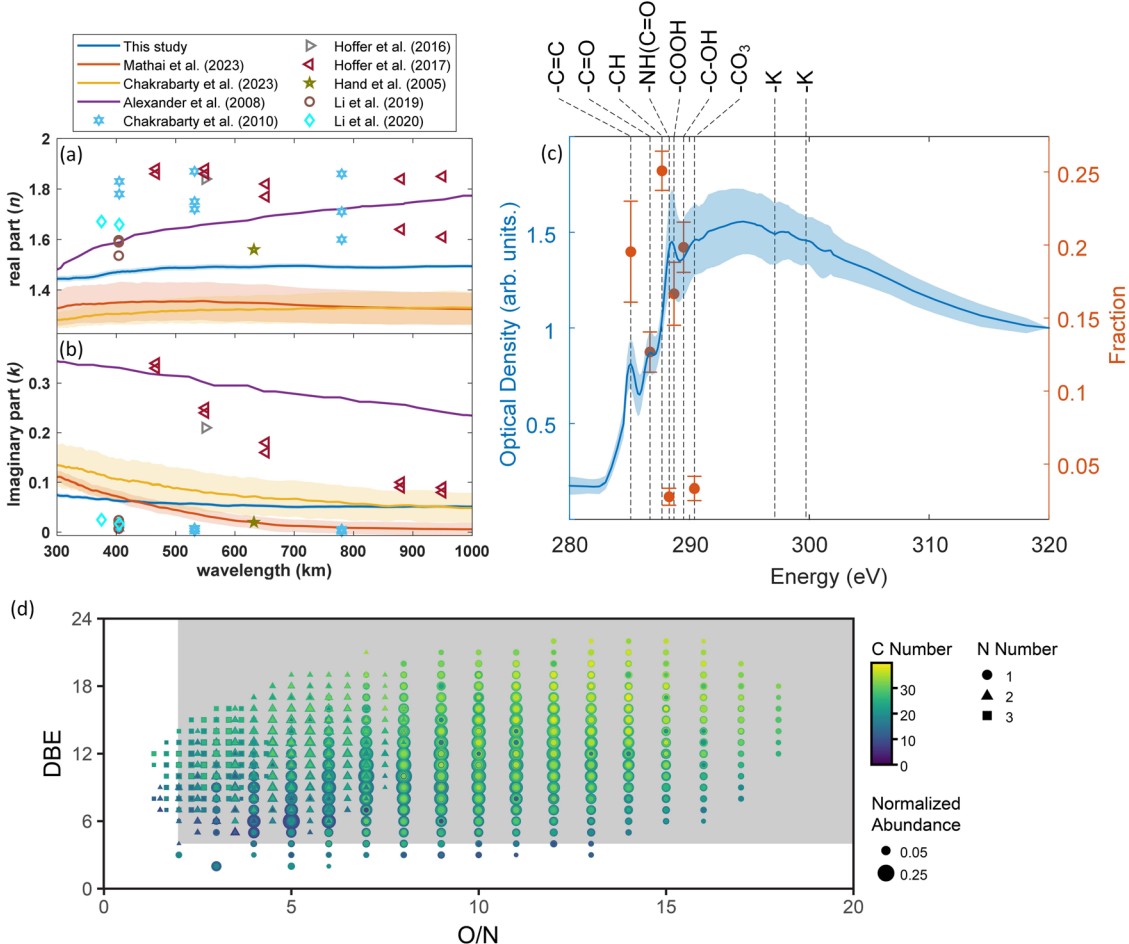

**Fig. 2 | Refractive index, carbon chemical bonding, and molecular composition of solid-state strongly absorptive brown carbon.** Mean (**a**) real part ($n$) and (**b**) imaginary part ($k$) of RIs against wavelength for solid S-BrC particles from this study and literature[7,10,12–18]. Shaded areas represent uncertainties. **c** Averaged STXM/NEXAFS spectra of individual solid S-BrC particles (left $y$-axis) and the averaged relative abundances of seven functional groups (C=C, C=O, -CH, -NH(C=O), -COOH, -C-OH, and -CO$_3$ (right $y$-axis)). STXM spectra for solid S-BrC show high -C=C, -COOH, and C-OH contributions. The shaded area in (**a**–**c**) represents measurement uncertainties as one standard deviation. **d** Double bond equivalent (DBE) as a function of the oxygen to nitrogen ratio (O/N) for organo-nitrate molecular formulae (CHNO compounds) from September 5, 2017, wildfire smoke samples analyzed by 21-T FT-ICR MS. The shaded area represents organo-nitrates, including nitrophenols, that account for 98% of all detected CHNO molecular formulae.

S-BrC, suggests the complex chemical composition of solid S-BrC. The abundance of the CNOK fraction (~11%) supports that the samples originated from wildfires[29]. This is consistent with the Hybrid Single-Particle Lagrangian Integrated Trajectory (HYSPLIT) model and fire location and intensity data, which show that air masses were influenced by wildfires in western Montana (Fig. S3).

Moreover, the size, shape, mass spectra, and volatility of over 30,000 individual aerosol particles were characterized in situ using the single-particle mass spectrometer (miniSPLAT)[30]. MiniSPLAT measurements of aerosol morphology and volatility indicated that most particles (>99%) were solid S-BrC because they were spherical and retained ~90% of their volume after 24 h of evaporation at room temperature[31]. Figure 1c shows an average miniSPLAT mass spectrum of all particles characterized over September 6 and 7. MiniSPLAT mass spectra exhibit mass spectral peaks characteristic of biomass-burning particles (K/K$_3$SO$_4$) ($m/z$ = 39 and 213) and indicate the presence of organic carbon (OC), including oxidized fragments, e.g., CO, C$_2$H$_3$O, CO$_2$, ($m/z$ = 28, 43, and 44), elemental carbon (EC) ($m/z$ = 12, 24, 36), polycyclic aromatic hydrocarbons (PAHs), sulfates, and organoni-trates. Based on single-particle mass spectra, these solid S-BrC can be classified into eight classes (Fig. S1, Section S1), suggesting the complex chemical nature of solid S-BrC. The single-particle mass spectra

analysis indicates particle-to-particle variability, mainly due to differ-ent mixing ratios of EC, PAHs, organics, and sulfate. This can affect variability in single-particle hygroscopicity and optical properties.

## Optical properties and chemical composition of solid S-BrC particles

To understand the direct climate effects of solid S-BrC, we probed the RI of 40 single solid S-BrC particles without any coating or inclusions as a function of wavelength (300–1000 nm) using electron energy-loss spectroscopy coupled to scanning transmission electron microscopy (EELS/STEM). As shown in Fig. 2a, the retrieved real part of solid S-BrC particle RI ($n$) increased from 1.44 to 1.48 from 200–500 nm while it was relatively constrained and wavelength-independent from 500–1000 nm (reported value of ~1.49). Conversely, the spectrum of retrieved $k$ exhibits a strong wavelength dependence, and the wave-length dependence of $k$ ($w$, $k(\lambda) = a\lambda^{-w}$) is 0.49 ± 0.5. The average $k_{550}$ is 0.056 ± 0.003, approximately an order of magnitude lower than BC[32]. The reported $w$ and $k_{550}$ fall in the S-BrC class, as suggested by Saleh[33].

Figure 2a, b shows the literature-reported RI of solid S-BrC[7,10,12–18]. The high variable RI might contribute to the complex chemical com-position and evaluation of their properties during atmospheric aging and transport, as well as bias from the bulk measurements and

analytical techniques[6,7,10–18,34]. As shown in Fig. 2a, b, our $n_{550}$ and $k_{550}$ are about 10% and 70% lower than those described in Alexander et al.[18], respectively. This discrepancy might be due to our advanced STEM's lower electron acceleration voltage (80 kV vs. 120 kV). This lower electron acceleration voltage results in lower Cherenkov radiation effects and electron beam-induced knock-on damage, which can overestimate the $k$[18,35,36]. Therefore, our optical properties of solid S-BrC can improve the uncertainties of solid S-BrC optical properties in the literature due to measurement limitations. Compared with two other recent studies[7,12], our $n_{550}$ is ~10% higher, and $k_{550}$ is ~73% higher than Mathai et al.[12] but ~29% lower than Chakrabarty et al.[7]. This variability can be attributed to differences in wildfire sources and chemical properties. The major component of our sample was western Montana wildfire smoke, which transported around 500 miles. In contrast, Mathai et al.[12] studied long-range transported smoke, and Chakrabarty et al.[7] studied local wildfire smoke. Thus, different transport distances and, consequently, the various extents of transformative atmospheric processes may explain variabilities in the optical properties of solid S-BrC.

In addition, the differences between optical properties reported in this study and in the literature can be attributed to the diversity in the solid S-BrC chemical composition[37]. MiniSPLAT shows particle-to-particle variability in solid S-BrC due to different mixing states of light-absorbing species, which might result in differences in their optical properties. Moreover, we used scanning transmission X-ray microscopy and near-edge X-ray absorption fine structure (STXM/NEXAFS) to probe the contribution of carbon functional groups of 67 particles. Figure 2c shows the averaged carbon K-edge spectrum of individual solid S-BrC, and the most intense peaks for these selected individual solid S-BrC particles are -COH ($0.20 \pm 0.02$), and -C=C (sp2) ($0.20 \pm 0.01$) peaks, followed by -COOH ($0.17 \pm 0.02$), which is consistent with literature-reported NEXAFS spectrum of solid S-BrC[8,12,38,39]. The high sp2 fraction in wildfire OA is associated with strong light absorption of solid S-BrC[32]. The range of the sp2 fraction (0.03–0.34) might explain the variation in $k$ due to the difference in the amount of delocalization in the molecule.

The diversity in molecular composition may shed light on the variation of $k$. The molecular composition of wildfire aerosols was analyzed by a 21-Tesla Fourier Transform-Ion Cyclotron Resonance mass spectrometer (21-T FT-ICR MS), showing that 50% of the 9841 total assigned organic molecular formulas were organonitrate compounds (Fig. 2d), which has been identified as an important

component in solid S-BrC[40]. 98% of detected organonitrates have molecular formulas consistent with nitroaromatics (C ≥ 6, double bond equivalent (DBE) ≥ 4, and O/N ≥ 2). Nitroaromatics are critical species in solid S-BrCs because they enhance light absorption in BrC at short wavelengths[41,42]. Furthermore, compounds with O/N ≥ 2 and up to four phenol ($C_6H_5O$) or two nitrophenol ($C_6H_5NO_3$) functionalities based on Kendrick mass defects were commonly detected here in the components of BrC wildfire smoke samples[41,43–46], providing a molecular basis for the light-absorbing nature of our solid S-BrC samples.

To conclude, our study comprehensively analyzed single solid S-BrC RI and chemical composition to investigate the association between light absorption properties of solid S-BrC, sp2 fraction per particle, and the fraction of organonitrate compounds. Future studies should focus on molecular composition data related to RI from identical individual particles to better understand the facts that affect solid S-BrC's strong light-absorption properties.

### Solid S-BrC absorption aerosol optical depth model sensitivity study

The highly variable RI of solid S-BrC in the literature can lead to significant uncertainties in predicted climate effects. We utilized the WRF-Chem regional model to perform a sensitivity study of RI on the absorption aerosol optical depth (AAOD) in the Pacific Northwest region from August 11 at 1:00 UTC to August 15 at 0:00 UTC, 2018. We show the percentage difference in the AAOD map from August 14 at 19:00 to 23:00 UTC, 2018. We utilized this period because of an existing model domain setup. Moreover, that period had solid S-BrC-rich wildfire smoke events where solid S-BrC's carbon K-edge spectrum (Fig. S5) and molecular formula similar to those in the 2017 study period[26,31], suggesting they have similar properties. We performed a sensitivity analysis of the impact of RI on AAOD by using RI reported by our measurements (Scenario 1), average RI during August 13 and 14, 2018, reported from the local Aerosol Robotic Network (AERONET) site (46.3 N, 119.3 W, Scenario 2, representing the lower bound) and RI from Alexander et al.[18] (Scenario 3, representing the upper bound). We chose AERONET because it is commonly used for model input to predict aerosol climate effects[47]. Figure 3a shows that using AERONET-reported RI resulted in around 50% lower AAOD in wildfire-influenced regions and 56% lower AAOD at the sampling site compared with Scenario 1. Contrary, the percentage difference of AAOD between Scenarios 1 and 3 shows that using RI from Alexander et al.[18] can lead to more than 200% higher AAOD at the sampling site (Fig. 3b). The high

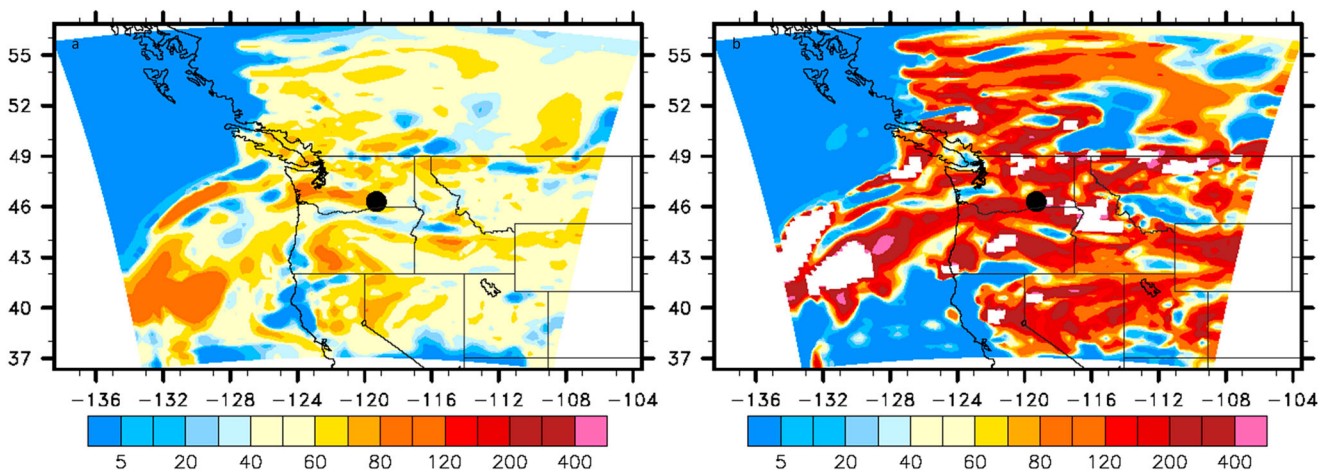

**Fig. 3 | WRF-Chem simulation for columnar absorption aerosol optical depth (AAOD) percentage differences. a** Percentage difference between Scenario 1 (RI from this study) and Scenario 2 (RI from AERONET) ([AAOD$_{scenario1}$-AAOD$_{scenario2}$]/ AAOD$_{scenario1}$*100) and that between (**b**) Scenario 1 and Scenario 3 (using RI from

Alexander et al.[18]) ([AAOD$_{scenario3}$-AAOD$_{scenario1}$]/ AAOD$_{scenario1}$*100) on August 14, 2018, averaged over 19:00–23:00 UTC. Markers represent the sampling site (46.3°N, 119.3°W). Color bars represent percentage differences. White area (**b**) is AAOD ratio greater than 500% for visual clarity.

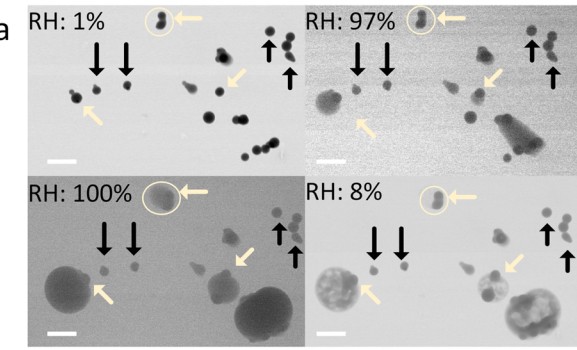

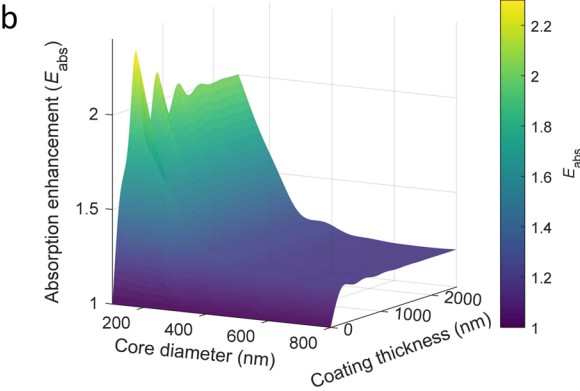

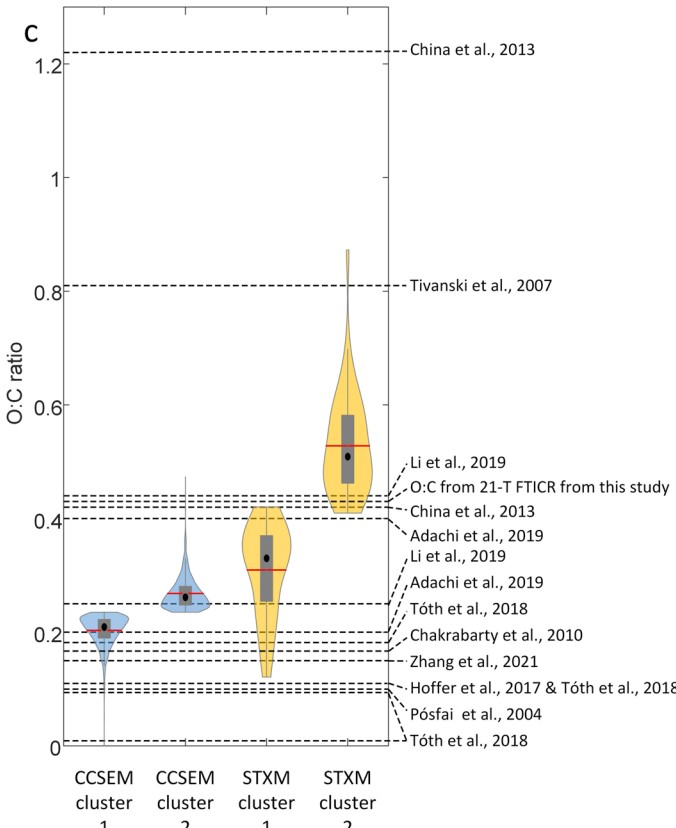

**Fig. 4 | Water uptake by solid S-BrC, lensing enhancement of solid S-BrC light absorption properties, and oxygen-to-carbon ratio for solid-state strongly absorptive brown carbon. a** Solid S-BrC water uptake experiment at 5 °C shows that some solid S-BrC did not uptake water (examples indicated by black arrows), and some are hydrophilic (examples indicated by light yellow arrows). Moreover, these solid S-BrC, which can uptake water, do not dissolve in water and form a water coating at high relative humidity (RH) conditions. The solid S-BrC selected by the light yellow cycles are solid S-BrC with thin organic coatings and can uptake water. The scale bar is 1 μm. **b** Lensing enhancement ($E_{abs}$) of solid S-BrC cores

(diameters from 100 to 800 nm, $RI_{solid\ S\text{-}BrC,550} = 1.49 + 0.056i$) coated with water (0–2500 nm thickness, $RI_{water,550} = 1.33 + 0i$) at 550 nm ($E_{abs,water}$), which can vary between 1.004 and 2.851 (see Section S3). The $E_{abs}$ is calculated as absorption cross-section ($\sigma_{abs}$) of the water-coated solid S-BrC particles ($\sigma_{abs,solid\ S\text{-}BrC,water}$) divided by $\sigma_{abs}$ of the solid S-BrC cores ($\sigma_{abs,\ solid\ S\text{-}BrC}$). **c** O/C elemental ratio from this study and literature[4,8,10,13,16,20,39,53,54]. We also reference this study's O:C ratio using 21-T FTICR MS data. The red lines indicate the means, the black dots are the medians, the gray rectangles are the interquartile ranges, the gray vertical lines are the 95% confidence intervals, and the violin-shaded areas show the data distribution.

variation in AAOD highlights the importance of improving the parameterization of the optical properties of solid S-BrC in the models to reduce the uncertainties in wildfire aerosol climate effects.

## Absorption enhancement of solid S-BrC due to water and organic coating

Besides the direct climate effects, there is limited knowledge about solid S-BrC indirect effects. Here, we conducted water-uptake experiments in an environmental SEM at 5 °C and analyzed more than 200 solid S-BrC particles. As shown in Fig. 4a, ~50% of the solid S-BrC did not uptake water (indicated by black arrows) even at >99% RH (additional images in Fig. S6). However, the rest of the solid S-BrC uptake water at RH above 97% (indicated by light yellow arrows, hereafter named hygroscopic solid S-BrC), suggesting a hygroscopic behavior and might potentially serve as cloud condensation nuclei (CCN) under supersaturated condition. Based on the aerosol size distribution[31] and MOUDI impactor collection efficiency[48], and assuming 50% of solid S-BrC particles are hygroscopic and will also activate into droplets at supersaturation conditions, we estimated that these hygroscopic solid S-BrC could result in CCN number concentrations $3692.2 ± 952.1$ cm$^{-3}$ between 131 and 445.1 nm particle size range during the sampling period. These estimates are comparable with modeled CCN in southeast Atlantic biomass-burning aerosol-dominated region[49]. Future studies are needed to better understand the contribution of hygroscopic solid S-BrC to CCN.

It should be noted that hygroscopic solid S-BrC (indicated by light yellow arrows) is still visible at RH > 97%, suggesting that most of the volume of solid S-BrC is hydrophobic. Moreover, this water forms a shell on hygroscopic solid S-BrC. Although these particles did not form core-shell morphology when collected on the TEM grids, we still cannot eliminate the potential for water coatings to fully cover the airborne hygroscopic solid S-BrC due to water surface tension. Moreover, after dehydration, we found that the hygroscopic solid S-BrC did not have obvious deformation. There was a residual surrounding the solid S-BrC (Fig. 4a). Our CCSEM-EDX (Fig. S7), STXM/NEXAFS Carbon speciation maps of solid S-BrC particles (Fig. S8), and the STEM/EDX Elemental map (Fig. S9) only show a negligible amount of inorganic elemental percentage ($1.3 ± 1.3\%$) and inorganic volume fraction ($0.04 ± 0.09$) in individual particles, respectively. Thus, we hypothesize that those residuals may have resulted from a thin, hygroscopic organic layer on the surface of the hydrophilic solid S-BrC particles that dissolved in water and then remained on the substrate surface after water evaporation. This hydrophilic organic layer may have resulted from condensation and deposition of polar organics and surface oxidation.

Some hygroscopic solid S-BrC particles have an organic coating (indicated by light yellow cycles), implying that hygroscopic organic coating can promote the water uptake of solid S-BrC. These observations support that solid S-BrC-water interaction is preferential, which could be attributed to the chemical heterogeneity of individual solid

S-BrC as suggested by single particle analysis. Adachi et al.[23] also reported an organic coating on solid S-BrC. These organic coatings might be formed by semi-volatile secondary organic vapor condensing on the solid S-BrC[23,50,51] and potentially be secondary BrC due to gas phase oxidation or heterogeneous oxidation after condensing on the solid S-BrC[1]. Moreover, this hygroscopic organic coating may have resulted from surface oxidation. Ijaz et al.[26] reported that using EELS maps, solid S-BrC in the Pacific Northwest in August 2018 had an oxygen outer layer. This conjecture is supported by Hand et al.[15] and Tivanski et al.[39], which reported that solid S-BrC has oxygenated outer layers, and China et al.[20] reported two types of solid S-BrC with different electronic darkness levels due to various degrees of surface oxidation. Therefore, these studies provide evidence of two types of solid S-BrC. However, the link between different types of solid S-BrC and their hygroscopicity is still unrevealed.

Because the hygroscopicity of organics can be predicted based on the O:C ratio[52], we retrieved the O:C ratio of individual solid S-BrC to develop a threshold to identify hygroscopic solid S-BrC particles. We also reference the O:C ratio from 21-T FT-ICR MS data. We compare this with the O:C ratio in the literature in Fig. 4c, which overlaps with our results but is highly variable due to diversity in analytical methods, solid S-BrC sources, and oxidation levels[4,8,10,13,20,39,53,54]. Furthermore, we utilized $k$-means clustering[55] to group solid S-BrC particles based on their O:C ratio (cluster 1: low O:C ratio, cluster 2: high O:C ratio) for both analytical techniques (see Section S2). We suggest that the hydrophobic solid S-BrC belongs to cluster 1 and solid S-BrC with hygroscopic coatings belongs to cluster 2 since organics with higher O:C ratios are more hygroscopic[52,56]. Therefore, we propose thresholds where individual solid S-BrC might be hydrophilic with an O:C ratio greater than 0.25 and 0.45 based on EDX and STXM, respectively.

The water shell on solid S-BrC can lead to lensing enhancement, increasing the solid S-BrC's light-absorption properties and warming effect and counterbalancing the cooling effect of solid S-BrC-containing clouds[57]. We utilized the model developed by Bond et al.[24] to estimate the absorption enhancement ($E_{abs}$) at 550 nm wavelength of solid S-BrC with diameters of 100–800 nm and water coating thicknesses between 0 and 2.5 μm, using our measured refractive index (RI) of solid S-BrC at 550 nm (Fig. 4b). We assume the solid S-BrC core is located at the center after uptaking water since solid S-BrC found in pyrocumulonimbus cloud droplets can located at the center of droplets[23]. We acknowledge this assumption might overlook the effects of the possibility that the core might not located at the center, which is worth future investigation. As shown in Fig. 4b, water shell can enhance the light-absorption of solid S-BrC by up to a factor of 2.3.

Besides the water shell, we observed organic coating on the solid S-BrC (Fig. S10). We might underestimate the fraction of organic-coated solid S-BrC based on microscopy imaging because these coatings might be evaporated in the high vacuum chamber. Considering this caveat, our finding suggests that solid S-BrC can act as a seed for secondary organic. Moreover, multiple studies have shown that secondary organic aerosols (SOA) can be light-absorbing[58,59]. Thus, these light-absorbing SOA can be coated on the solid S-BrC and cause even higher absorption enhancement than water coating for the entire particle via both lensing effect and absorption by the coating[25]. We estimated the absorption enhancement loss (the difference between lensing enhancement with a clear coating and with a light-absorbing coating)[25] and the value to be up to a factor of 1.3 (Fig. S11). These results confirm that the organic coating can potentially increase the contribution of solid S-BrC to climate warming[60]. Further study is needed to better quantify the fraction of organic coated solid S-BrC and the light-absorption enhancement due to light-absorbing SOA coating.

## Discussion

Our study suggests solid S-BrC can be a major component in some wildfire smoke. Without accurately representing it, climate models might underestimate the warming effect of wildfire smoke. Moreover, our finding suggests that ~50% of solid S-BrC are hygroscopic and can act as CCN at high RH environments, leading to cloud-heating effects[61]. Thus, considering hydroscopic solid S-BrC in models might improve the predicted aerosol indirect climate effects. Previous studies primarily focus on the lensing enhancement of light-absorbing and Directive Radiative Forcing (DRF) of soot[62,63]. Our study established a previously unrecognized concept by showing that water coating on solid BrC can cause lensing enhancement up to 2.3 at $k_{550}$. Since the WRF-Chem model does not include parameterization of lensing enhancement of solid S-BrC, we used the theoretical calculation to estimate the top-of-the-atmosphere DRF[64]. It shows that a 200-nm thick clear coating can lead to ~43% enhancement in 200 nm diameter solid S-BrC, enhancing directive radiative forcing at 550 nm ($E_{DRF,550}$) (see section S4). Moreover, a light-absorbing organic coating can increase the lensing enhancement, leading to ~240% $E_{DRF,550}$ of 200 nm diameter solid S-BrC with 200 nm thick coating. It should be noted that the different RI of coating, coating thickness, and core size can lead to large variations in the results. Thus, future studies to better understand the climate effects of coated solid S-BrC are necessary to parameterize our findings in climate models to reduce the discrepancy between measurements and improve the model's accuracy. These findings should be parameterized in climate models to reduce the discrepancy between measurements and improve the model's accuracy.

## Methods

### Biomass-burning particle collection

Wildfire aerosols were collected daily from 9 AM to 2 PM with 50% duty cycle (30 min on and 30 min off) from September 5–6, 2017, and August 9–14, 2018, local time at the Atmospheric Measurement Laboratory in Richland, WA (46.340844 N, 119.278110 W). In this study, all analysis focuses on samples collected from September 5–6, 2017, and ambient temperature and relative humidity during the sampling period are shown in Fig. S12. Details about samples collected from August 9–14, 2018, have been discussed in Ijaz et al.[26]. Samples for individual particle analysis were collected on Carbon B-film TEM grids (Ted Pella, Inc.) using a MOUDI impactor (model 110-R, MSP, INC) at a flow rate of ~30 LPM. Samples were collected daily on PTFE filters at the same location for offline mass spectrometry analysis. The 48-hour HYSPLIT[65] combined with the fire map based on the observation from the Visible Infrared Imaging Radiometer Suite aboard the Suomi National Polar-orbiting Partnership (S-NPP) satellite[66] (Fig. S3) indicated that the wildfire in west Montana State was the primary source of Pacific Northwest regional smoke.

### Single-particle imaging and analysis

We imaged collected particles at both horizontal and a 75° tilt angle to visualize the deformation that particles undergo upon their impaction on the substrate[27]. Collected particle samples were analyzed with CCSEM-EDX and STXM/NEXAFS spectroscopy[67,68]. CCSEM-EDX is an environmental SEM (ESEM) equipped with an FEI Quanta digital field emission gun operated at 20 kV and 480 pA current to probe size (area equivalent diameter), shape, and morphology. The EDX spectrometer (EDAX, Inc.) interfaced with CCSEM was used to probe the chemical composition of individual particles. This study quantified the relative atomic ratios of 15 elements (C, N, O, Na, Mg, Al, Si, P, S, Cl, K, Ca, Mn, Fe, and Zn). The relative atomic fraction of these elements in each particle was used to classify the particle type (see the Supporting Information). This study analyzed more than 100,000 particles. Based on a relative atomic percentage of individual particles, we classified

each particle as carbonaceous (CNO), CNO with sulfate (CNOS), CNO with K (CNOK), sodium-rich (Na-rich), dust, bioaerosol, or others (Fig. S4). Samples with a high abundance of solid S-BrC were selected for further analysis via STEM.

We retrieved RI values using the same method as Veghte et al.[69], and the STEM/EELS setup was the same as that used by Mathai et al[12]. Individual solid S-BrC with a diameter range from 50–200 nm were analyzed with aberration-corrected electron energy-loss spectroscopy (Gatan) coupled to the scanning transmission electron microscope (EELS/STEM) operated at 80 kV. The energy dispersion of the EELS monochromator is 0.025 eV/channel. The collection angle is 45.5 mrad, and the particle thickness was estimated as the diameter of particles. Infrared spectra for individual solid S-BrC at wavelengths from 300 to 1000 nm were retrieved from the single scattering distributions in the low-loss region (0–10 eV) of the EELS spectrum after removing the zero-loss peak. We performed the Kramers−Kronig analysis on extracted single scattering distributions to retrieve the RI[70].

The electron beam (20 kV for ESEM and 80 kV for STEM) has the potential to modify the particles, and volatile species might be lost in the high vacuum of the ESEM (-2×10$^{-6}$ torr) and STEM chamber (-2×10$^{-9}$ torr)[69]. These caveats should be considered when interpreting TEM results.

Water-uptake experiments were conducted inside the ESEM using a Peltier stage. The temperature was maintained at 278 K, and the RH was gradually increased from 1% to 99% at an interval of 5–10%, then reduced to <10%. Images were captured 2 min after the water vapor pressure reached the set point to ensure the system reached a steady state. We utilized the STEM detector to acquire high-resolution ESEM images. More than 200 particles were analyzed. More water-uptake results can be found in Fig. S6. Because the titled SEM images show tar balls dominating, these round dark particles are tar balls.

The STXM/NEXAFS spectroscopy at beamline 5.3.2 (Advanced Light Source, the Lawrence Berkley National Laboratory) was utilized to probe the X-ray spectrum and chemical bonding of carbon functional groups of samples. The contribution of functional groups was analyzed by performing spectral deconvolution and nonlinear least-squares fitting of seven peaks (C=C, C=O, -CH, -NH(C=O), -COOH, -C-OH, and -CO$_3$) associated with specific functionality to determine the area under each peak[71–73]. The masses of C and O were calculated based on the difference between the post- and pre-K-edge optical depth for specific element times, where the projection area of the particle was divided by the difference between the mass absorption coefficients before and after the absorption edge.

To calculate the O: C ratios, we used the C and O elemental ratios of CNO, CNOS, and CNOK particles from CCSEM/EDX and the carbon and oxygen maps of solid S-BrC from STXM/NEXAFS.

**21-Tesla Fourier transform ion cyclotron resonance mass spectrometer analysis**
A small part of the September 5, 2017, PTFE filter was extracted in 100% acetonitrile by sonicating for 30 min, followed by filtration through a 0.2 μm PTFE syringe filter. The filtrate was diluted with a 70:30 Acetonitrile:H$_2$O solution for mass spectral analysis on a custom-built 21-T FT-ICR MS at the Environmental Molecular Sciences Laboratory, Pacific Northwest National Laboratory, Richland, Washington, USA[74]. Data acquisition details were previously described in Ijaz et al.[75]. Exact masses were exported to MFAssignR[76] for formula assignment using the previously described method[75]. Briefly, noise was estimated in the mass lists using the KMDNoise(), and monoisotopic and polyisotopic peaks were separated with IsoFiltR(). Preliminary formulae were assigned with C, H, and O only using MFAssignCHO(), and suitable recalibrants were selected from these and used in Recal() for internal recalibration. Molecular formulae were then assigned using MFAssign() as follows: C$_c$H$_h$O$_o$N$_{0-3}$S$_{0-2}$$^{13}$C$_{0-2}$$^{34}$S$_{0-1}$; -13 ≤ DBE-O ≤ 20; 0 ≤ O/C ≤ 2.0; and 0.3 ≤ H/C ≤ 2.5, with a maximum permissible error of 0.5 ppm. Kendrick mass defect analysis was performed to estimate the functional groups present. Structural information of the molecules identified was inferred by calculating the DBE.

**WRF-Chem simulation**
The regional Weather Research and Forecasting Model coupled with chemistry (WRF-Chem 4.2)[77–80] was used to simulate biomass-burning aerosols from August 8 to August 15 00:00 UTC, 2018, in a 2187 × 2187 km domain with a spatial resolution of 27 × 27 km in the Pacific Northwest region for the domain shown in Fig. 3 with 72 vertical levels, with the first 3 days used for model spinup. We include primary biomass-burning emissions for gases and aerosols from the 2014 Quick Fire Emissions Database (Darmenov & da Silva, 2015) version 2.5. Biogenic volatile organic compounds emissions are derived from the latest version of the Model of Emissions of Gases and Aerosols from Nature (MEGAN v2.1), recently coupled within the community land model CLM4 (CLM version 4.0) in WRF-Chem[81]. The CAM-chem global model[82] provided initial and boundary conditions for trace gases and aerosols. The Global Forecast System model provides the meteorological initial and boundary conditions. Meteorological and chemical conditions were spun up for 72 hours using WRF-Chem, followed by the 24-h simulation for a given day of interest. Gas-phase chemistry is based on the Statewide Air Pollution Research Center (SAPRC-99) mechanism, which includes 211 reactions of 74 gas-phase species, 18 of which are free radicals. Inorganic aerosol chemistry, and the evolution of aerosol size distribution and microphysics in WRF-Chem are represented by the Model for Simulating Aerosol Interactions and Chemistry (MOSAIC)[83]. Aerosol species in MOSAIC include sulfate, nitrate, ammonium, other inorganics, EC, organic aerosols, and aerosol water. Aerosols are assumed to be internally mixed and are represented by eight sections with dry particle diameter ranges of 0.039–0.078, 0.078–0.156, 0.156–0.312, 0.312–0.624, 0.624–1.25, 1.25–2.5, 2.5–5.0, and 5–10 μm. Hourly aerosol and trace gas emissions from sources other than biomass-burning and biogenic emissions are derived from the United States Environmental Protection Agency's 2014 National Emissions Inventory (NEI2014). We represent SOA formed due to the oxidation of biogenic, anthropogenic, and biomass-burning organic gases using our previously documented volatility basis set approach[80] that has shown good agreement with field observations. WRF-Chem calculated aerosol optical depth (AOD) and single scattering albedo (SSA) at each vertical layer using the aerosol number density retrieved from the aerosol size distribution[31] and aerosol RI using the Mie theory[84]. The AAOD is calculated as AAOD = AOD × (1-SSA) at each layer and then integrated vertically to get the columnar AAOD. The supporting Information (Tables S2–4) provides further details of the WRF-Chem model configuration. Currently, the WRF-Chem simulation does not include lensing effects.

## Data availability
Data are reported in tabular form in the supplementary information. Experimental data have been deposited in open-access data repository (https://zenodo.org/records/13798865)[85]. Data supporting the findings of this manuscript are also available from the corresponding authors upon request.

## Code availability
We used the Community Model WRF-Chem available online. Model outputs from WRF-Chem that are used to generate figures in this study are available from the corresponding author on request.

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

## Acknowledgements

A portion of this research was performed on a project award (https://doi.org/10.46936/lser.proj.2019.50795/60000103) from the Environmental Molecular Sciences Laboratory, a Department of Energy Office of Science User Facility sponsored by the Biological and Environmental Research (BER) Program. PNNL is operated for the Department of Energy

by Battelle Memorial Institute under contract DE-AC06-76RL0. STXM/NEXAFS analysis at beamline 5.3.2 of the Advanced Light Source at Lawrence Berkeley National Laboratory is supported by the Department of Energy, Office of Science, Basic Energy Sciences program. Manish Shrivastava was supported by the U.S. DOE, Office of Science, BER Atmospheric System Research (ASR) program and the DOE BER Early Career Research program. Gourihar Kulkarni, Jerome Fast, Kaitlyn Suski, and Alla Zelenyuk were supported by the DOE's Atmospheric System Research (ASR) program.

## Author contributions

S.C. and Z.C., conceived and designed the research. Z.C. led the manuscript writing. L.K., D.V. and S.C. performed STEM/EELS experiments and analysis. Z.C., N.N.L., J.W., and S.C. collected and performed CCSEM-EDX and STXM/NEXAFS analysis. K.T. performed water uptake experiments. A.I., G.W.V., W.K., and L.R.M. contributed to mass spectrometry analysis and data interpretation. K.S. and A.Z. performed mini-iSPLAT sampling and data interpretation. G.K. and S.C. contributed to the sample collection and analyze meteorological data. Z.C. performed the optical calculations. M.S., L.K.B., and J.D.F. set up the regional domain for WFR-Chem, and M.S. ran the model and analyzed model data. All co-authors contributed to review the manuscript and response to reviewers' comments.

## Competing interests

The authors declare no competing interests.
