## [Transparent Peer Review file · Nature Communications]

Enhanced light absorption for solid-state brown carbon from wildfires due to organic and water coatings

Corresponding Author: Dr Swarup China

Version 0:

Reviewer comments:

Reviewer #1

(Remarks to the Author)

The study reported solid-state strongly absorptive brown carbon (S-BrC) from one wildfire. They used SEM, TEM and other techniques to quantify the morphology, mixing state, compositions of S-BrC after one transported distance from wildfire. The authors nicely used the EELS to quantify the absorption of S-BrC. Then they found that S-BrC particles could be hydrophilic based their experiments. Although there were many studies revealing the tarballs in the troposphere in the world, the study seemingly had one advancement to better understand their optical and hygroscopicity. The study could be made an important improvement to understand the Tarballs from wildfire. Basically, I agree that the high quality of this paper can go to Nature Communications. However, I also raised some points which should be better answered.

Abstract:

- 1) "The mean refractive index of $1.49+0.059i$ at 550 nm." Could the authors check the number of 0.059, why I could not find it in main context. Basically, the number should be existing in context before I can go to abstract. Please make sure the number is right. In fact, the refractive index is not surprising for me, because there were some studies reported these number.
- 2) L50-52, one recent study well reviewed the absorptive particles. *npj Climate and Atmospheric Science* 2024, 7(1): 65.
- 3) L125, please K550 is 0.056 instead of 0.059 in abstract.
- 4) L151-158, Basically, one study well reported the BrC contain organic nitrate based on one special technique Yuan et al., *Environ. Sci. Technol. Lett.* 2021, 8(1): 16-23. The authors can cite this to support conclusion from bulk samples.
- 5) L199-203. The Figure 4a, could be key in this study. I would like to ask to check whether the tarballs indicated by the light yellow cycles internally mixed with minor secondary species such as sulfate or K-sulfate. Here the Figure 4a is not clear and high enough magnifications to display the clear structure. I am afraid that the author emphasize the part due to your technique instead of the fact. That could make wrong side.
Of course, I finally agree with your conclusions. In fact, most of particles are more or less coated by secondary species or the condensed organic species. As the *Proc. Natl. Acad. Sci. U. S. A.* 116, 19336–19341, they found these particles dynamic changed of the surface depending the environmental condition. Following the transport distance, these tarballs mostly internally mixed with others. Finally, they can be CCN. However, I just worried about the hygroscopicity is not from the surface of the sole tarball.
- 6) Figure 4c, these references should be added in caption. Otherwise, the potential readers could not locate these references.
- 7) Finally, the title, why there is no inorganic. Inorganic in aged tarball also contribute the optical absorption. Please consider how to revise it and widely accepted by the readers. I think that current one might make mistake your work.

Reviewer #2

(Remarks to the Author)

In this paper, Cheng et al. have made several high quality measurements on solid-state strongly absorptive brown carbon (solid S-BrC) in terms of composition, optical properties, and hygroscopicity. Their measurements provide variable knowledge of S-BrC, which is an important factor of climate. Thus, I have no doubt that this work will contribute to atmospheric and climate science. On the other hand, as the journal aims to "represent important advances of significance to specialists within each field", I question whether the results of this paper reach that level. They found a significant number of S-BrC from a small set of samples (two-day sampling) and showed their compositions and optical properties. These results

agree with previous studies and strongly supports the previous observations using a state-of-the-art instrument. On the other hand, the results are not highly novel. In addition, although the results and discussions of the coating on S-BrC on the radiative forcing are novel, the discussions are not directly supported by their observations. Therefore, I cannot strongly support this paper being published in this multidisciplinary high impact journal.

Specific comments.

Line 69: "collected during the Pacific Northwest wildfire events on September 5 and 6, 2017" and line 403 "from September 5–6, 2017, and August 9–14, 2018,": It is unclear whether the samples were collected only in 2017 or in both 2017 and 2018. In either way, the sampling periods are limited and it is difficult to draw a general conclusion from the limited samples.

Line 79 "We manually identified particle types based on their morphology" Please define the morphological criteria. It will also affect the discussion of the hygroscopicity study in Fig. 4.

Line 130-133: "As shown in Figs. 2a and 2b, our n550 and k550 are about 10% and 70% lower than those described in Alexander et al., respectively. This discrepancy might be due to our advanced STEM's lower electron acceleration (80 kV vs. 120 kV), which results in lower Cherenkov radiation effects and electron beam-induced knock-on damage, which can overestimate the k" This discussion suggests that the detailed comparison of refractive indices from this study and those of Alexander et al in Fig. 3 is meaningless, since the difference is due to the uncertainties of the measurements.

Line 127-160: In the second paragraph of this section (line 127-139), the authors explain the difference in refractive indices due to different atmospheric processes. In the third and fourth paragraphs, they discuss the differences based on SP2 fractions and molecular compositions, respectively. Although all these processes may contribute to the refractive indices, a comprehensive discussion is helpful.

Line 174-177: "We show the percentage difference in the AOD map from August 14 at 19:00 to 23:00 UTC, 2018. We utilized this period because of an existing model domain setup and because that period had solid S-BrC-rich wildfire smoke events with similar molecular compositions to the 2017 study period." The authors explain why they chose a different time period between their observations and modeling; however, it is better to use the same time period to draw a solid conclusion.

Line 233-237: "We might underestimate the fraction of organic-coated solid S-BrC based on microscopy imaging because these coatings might be evaporated in the high vacuum chamber. Considering this caveat, our finding suggests that solid S-BrC can act as a seed for secondary organic aerosol formation and that the organic coating can potentially increase the contribution of solid S-BrC to climate warming." The discussion in the second sentence is based on a hypothesis and lacks evidence. In the abstract, they mention that "Furthermore, the organic coating on solid S-BrC can lead to even higher lensing enhancements than water," which is not supported by the evidence.

Fig. 4a and S5. "solid S-BrC that uptake water" Some of them look like they are deliquescing rather than absorbing water. This result may suggest that inorganic salt that attached to or internally mixed with the S-BrC has deliquesced. If the authors have considered this possibility, I suggest adding some discussion. At least, I found many sulfate in Fig. S1.

Fig. S4. Is the first question (Yes/No for Na>1%) correct?

Reviewer #3

(Remarks to the Author)

Review of "Enhance light absorption from solid-state brown carbon from wildfires due to organic and water coatings" by Cheng et al.

The study focused on a wildfire event on September 5 and 6, 2017 at a single location. The researchers conducted an extensive and comprehensive analysis of the wildfire smoke, offering a very detailed picture of the physical properties of particles from this wildfire event, including optical properties and some aspects of hygroscopicity. Most of the data and analysis are high quality and should be published. However, I wonder if the paper fits within Nature Communications' mandate. This aspect is up to the Editor.

The paper consists of four main sections: 1) prevalence of solid S-BrC in wildfire smoke aerosol, 2) optical properties and chemical composition of solid S-BrC particles, 3) sensitivity study of solid S-BrC absorption aerosol optical depth using model simulations, and 4) absorption enhancement due to water-solid S-BrC interactions and organic coatings. Sections 1-3 are not novel but still significant. Section 4 presents the most novel results, but the conclusions is less convincing than the other sections.

Detailed comments:

1) Clarity and context: I had to read the paper a couple of times to appreciate what is novel in the paper. Also, the presentation feels a little disjointed in places. For example, Sections 1-3 seem separate from Section 4. I think Sections 1-3 could be a separate paper.

2) Title: The title only highlights Section 4. I.e., the title does not represent all the content of the manuscript.

3) Environmental implications: The environmental implications only cover the results in Section 4. This is one reason for arguing that Sections 1-3 could be a separate paper.

4) Significance: the most novel aspect of the paper is the observation that approximately half of the solid S-BrC are hydrophobic, and half take up water at 97 % RH. Even this observation is not completely shocking since a) one previous paper by the authors observed two types of solid S-BrC in a previous field campaign, b) hydrophobic and hydrophilic solid S-BrC have been observed previously in separate studies, as pointed out by the authors, and c) a recent study observed solid S-BrC in cloud residuals (Adachi et al. Egusph. 2024) indicating that at least some solid S-BrC are hydrophilic. These previous studies may compromise the manuscript's significance.

5) Validity of the data: I have questions about the validity of the data interpretations and conclusions, especially the results in Section 4. See below.

6) Page 7, lines 198 to 201: "As shown in Fig. 4a, ~50% of the solid S-BrC did not uptake water (indicated by black arrows) even at >99% RH (additional images in Fig. S5). However, the rest of the solid S-BrC uptake water at RH above 97% to form a core-shell morphology with an aqueous shell (indicated by light yellow arrows, hereafter named hygroscopic solid S-BrC), suggesting a potential function as cloud condensation nuclei." Based on Fig. 4a, it doesn't look like the particles form core-shell morphology, contrary to the suggestions of the authors. It looks more like the solid S-BrC often remains at the interface of the water droplets. Also, one of the particles with a light yellow circle doesn't appear to take up any water (the particle with a light yellow circle at the top right-hand corner of the figure).

7) Page 7, lines 202-203: "Some hygroscopic solid S-BrC particles have an organic coating (indicated by light yellow circles), implying that hygroscopic organic coating can promote the water uptake of solid S-BrC." What is the proof of an organic coating? Is the proof the observation of a larger "residual" after evaporation? One of these particles with a light yellow circle did not leave a larger "residual" after evaporation. Also, in some cases, the residual may have been caused by a water-soluble particle nearby. Also, how can the authors differentiate between partial dissolution of the solid S-BrC and the presence of an organic coating? The solid S-BrC particles could partially or completely dissolve if enough water is added.

8) Page 8, lines 235-237: "Considering this caveat, our finding suggests that solid S-BrC can act as a seed for secondary organic aerosol formation and that the organic coating can potentially increase the contribution of solid S-BrC to climate warming." I don't see conclusive evidence for an organic coating since alternative explanations are also possible for the observations. So, I don't think the results provide convincing evidence that solid S-BrC can act as a seed for secondary organic aerosol formation.

9) Page 9, lines 254-256. "Our study established a previously unrecognized concept by showing that water coating on solid S-BrC can cause lensing enhancement up to 2.3 at k_{550} ." This assumes the solid S-BrC is in the center of the water droplets (i.e., core-shell), and the solid S-BrC doesn't dissolve in water. However, the images (Figure 4) suggest they will not form a core-shell structure, and some of the images indicate that the solid S-BrC may have partially dissolved. In the current experiments, the water uptake is less than what will happen in a cloud droplet. Perhaps in a cloud droplet, the solid S-BrC will completely dissolve.

10) The authors argue that some of the solid S-BrC have an organic coating. How does this affect their optical constants measurements? I guess the authors assumed the solid S-BrC did not have an organic coating when extracting the optical constants.

Version 1:

Reviewer comments:

Reviewer #1

(Remarks to the Author)

I carefully evaluate the response. I am happy on the revision. I basically go to accept this article in NC. Here I found one minor point.

L302-303, How did the author determine 240%. The authors should cite the sources.

Here I do want make challenge this number. The light absorbing organic coating might have shield effect instead of the light enhancement. How did the author find even bigger Eabs? How do you set up the optical refractive index of light absorbing organic? Please make more suitable implication here.

Reviewer #2

(Remarks to the Author)

The authors have considered the reviewers' comments and provided appropriate responses and discussion in the revised paper. I agree with the authors' responses and have no further comments. I hope that this paper makes a significant contribution to the understanding of brown carbon from wildfires.

Reviewer #3

(Remarks to the Author)

Second review of “enhanced light absorption for solid-state brown carbon from wildfires due to organic and water coatings”.

I thank the authors for taking my comments seriously. However, I still don't think the paper warrants publication in Nature – Communications, which publishes papers that represent “important advances of significance to specialists within the field”. The paper is a good one, but doesn't quite reach the standard of other papers from the same field that have been published in Nature – Communications.

Specific comments:

- 1) The clarity of the paper still needs to be improved, especially if the authors are considering a higher impact and more general journal. In several places it is still not clear what is new and what has already been done.
- 2) the number of samples still seems too small to make strong general comments.
- 3) Significance: In most cases, the results are not new. The new aspects do not seem significant enough for a higher impact more general journal.
- 4) Some of the implications are uncertain due to speculation. For example, the authors assume a core-shell model when discussing the effect of solid S-BrC on cloud droplets, but whether or not this is valid is not clear.

I do think the paper is a nice contribution to the field of atmospheric science and should be published. I also commend the authors for the large body of work.

We want to thank the reviewers for their constructive comments. Addressing those comments has improved the quality of the manuscript. Below, we list each reviewer's comment (regular font), followed by our response (indented, **bold font**), followed by corresponding changes in the revised manuscript (*indented, blue font*). RL and RSL represent the line number in the revised main manuscript and SI, respectively.

Reviewer #1 (Remarks to the Author):

The study reported solid-state strongly absorptive brown carbon (S-BrC) from one wildfire. They used SEM, TEM and other techniques to quantify the morphology, mixing state, compositions of S-BrC after one transported distance from wildfire. The authors nicely used the EELS to quantify the absorption of S-BrC. Then they found that S-BrC particles could be hydrophilic based their experiments. Although there were many studies revealing the tarballs in the troposphere in the world, the study seemly had one advancement to better understand their optical and hygroscopicity. The study could be make important improvement to understand the Tarballs from wildfire. Basically, I agree that the high quality of this paper can go to Nature Communications. However, I also raised some points which should be better answered.

We appreciate that the reviewer considers the high quality of the manuscript and deserving publication. As the reviewer suggested, we modified the highlights of the manuscript (please see below). Below are our responses to each comment:

Abstract:

1) "The mean refractive index of $1.49+0.059i$ at 550 nm." Could the authors checked the number of 0.059, why I could not find it in main context. Basically, the number should be existing in context before I can go to abstract. Please make sure the number is right. In fact, the refractive index is not surprising for me, because there were some studies reported these number.

We want to thank the reviewer for pointing this out. We have corrected the value in the abstract. It should be 0.056. Please see our revised text below:

RL35-37: "Here, we show that from a Pacific Northwest wildfire in September 2017, greater than 90% (by number) of particles were solid S-BrC with a mean refractive index of $1.49+0.056i$ at 550 nm."

2) L50-52, one recent study well reviewed the absorptive particles. npj Climate and Atmospheric Science 2024, 7(1): 65.

We thank the reviewer for sharing this reference. We have cited this paper in the revised manuscript.

RL51-53: "BC is typically considered the dominant light absorber,² but studies have reported that the wildfire smoke particles contain a special type of solid-state strongly absorptive BrC (solid S-BrC) commonly known as tar balls.³⁻⁵"

3) L125, please K550 is 0.056 instead of 0.059 in abstract.

This has been corrected in the abstract.

4) L151-158, Basically, one study well reported the BrC contain organic nitrate based on one special technique Yuan et al., Environ. Sci. Technol. Lett. 2021, 8(1): 16-23. The authors can cite this to support conclusion from bulk samples.

We appreciate the reviewer for sharing this relevant reference. We have added this reference and a relevant discussion to the discussion section in the revised manuscript.

RL160-164: “The molecular composition of wildfire aerosols was analyzed by a 21-Tesla Fourier Transform-Ion Cyclotron Resonance mass spectrometer (21-T FT-ICR MS), showing that 50% of the 9,841 total assigned organic molecular formulas were organonitrate compounds (Fig. 2d), which has been identified as an important component in solid S-BrC.³⁹”

5) L199-203. The Figure 4a, could be key in this study. I would like to ask to check whether the tarballs indicated by the light yellow cycles internally mixed with minor secondary species such as sulfate or K-sulfate. Here the Figure 4a is not clear and high enough magnifications to display the clear structure. I am afraid that the author emphasize the part due to your technique instead of the fact. That could make wrong side.

Of course, I finally agree with your conclusions. In fact, most of particles are more or less coated by secondary species or the condensed organic species. As the Proc. Natl. Acad. Sci. U. S. A.116, 19336–19341, they found these particles dynamic changed of the surface depending the environmental condition. Following the transport distance, these tarballs mostly internally mixed with others. Finally, they can be CCN. However, I just worried about the hygroscopicity is not from the surface of the sole tarball.

We appreciate the reviewer’s constructive comments. We agree with the reviewer that atmospheric-aged solid S-BrC can be internally mixed with hygroscopic species such as K and sulfate (see ref.¹⁻⁴). We also agree that our SEM images might not have high enough resolution to see the detailed internal structure of individual particles. However, we observed that most tar balls did not have inorganics based on CCSEM-EDX, STXM/NEXAFS, and TEM results. We acknowledge the potential effects of inorganic inclusions, but this might not be the case in our study since our CCSEM-EDX and STXM/NEXAFS only show a negligible amount of inorganic elemental percentage ($1.3\pm 1.3\%$) and inorganic volume fraction (0.04 ± 0.09) in individual particles, respectively. In addition, due to the hydrophobic nature of solid S-BrC, water might not be able to easily diffuse into the particle. Since we observe different hygroscopicity between different solid S-BrCs, we believe the K and S might not be the reason for that different hygroscopicity. Figure 4 a shows water (lighter gray, indicated by white arrow) coated on the Solid S-BrC (black, indicated by dark gray arrow) in 97%RH and 100%RH. This suggests that the Solid S-BrC does not dissolve in water after water uptake, and the water uptake happens on the surface of Solid S-BrC. However, we have not eliminated the trace amount of inorganics as a potential cause for the water uptake. We will add some discussion below:

RL225-237: “It should be noted that hygroscopic solid S-BrC (indicated by white cycles) is still visible at RH>97%, suggesting that most of the volume of solid S-BrC is non-hydrophobic. Moreover, this water forms a shell on hygroscopic solid S-BrC. Although these particles did not form core-shell morphology when collected on the TEM grids, we still cannot eliminate the potential for water coatings to fully cover the airborne hygroscopic solid S-BrC due to water surface tension. Moreover, after dehydration, we found that the hygroscopic solid S-BrC did not have obvious deformation. There was a residual surrounding the solid S-BrC (Fig. 4a). Our CCSEM-EDX (Fig. S7), the Scanning

Transmission Electron Microscopy Elemental map (Fig. S8) and STXM/NEXAFS Carbon speciation maps of solid S-BrC particles (Fig. S9) only show a negligible amount of inorganic elemental percentage ($1.3 \pm 1.3\%$) and inorganic volume fraction (0.04 ± 0.09) in individual particles, respectively. Thus, we hypothesize that those residuals may have resulted from a thin, hygroscopic organic layer on the surface of the hydrophilic solid S-BrC particles that dissolved in water and then remained on the substrate surface after water evaporation. This hydrophilic organic layer may have resulted from condensation and deposition of polar organics and surface oxidation.”

Figure S7. Elemental percentage of individual particles analyzed by CCSEM-EDX.

Figure S8. STXM/NEXAFS Carbon speciation maps of solid S-BrC particles.

Figure S9. Elemental map of solid S-BrC acquired by Scanning Transmission Electron Microscopy (Thermo Fisher, model Titan 80-300) (STEM)/EDX with energy-dispersive X-ray spectroscopy (Oxford Instruments).

6) Figure 4c, these references should be added in caption. Otherwise, the potential readers could not locate these references.

We thank the review's suggestion, and we have revised the caption as below:

RL176-178: “Fig. 2 | Refractive index, carbon chemical bonding, and molecular composition of solid S-BrC. Mean (a) real part (n) and (b) imaginary part (k) of RIs against wavelength for solid S-BrC particles from this study and literature.^{7,10,12–18”}

RL279-290: “Fig. 4. | Water uptake by solid S-BrC, lensing enhancement of solid S-BrC light absorption properties, and oxygen-to-carbon ratio for solid S-BrC. (a) Solid S-BrC water uptake experiment at 5°C shows that some solid S-BrC can uptake water (examples indicated by black arrows), and some are hydrophilic (examples indicated by light yellow arrows). Moreover, these solid S-BrC, which can uptake water, do not dissolve in water and form a core-shell morphology at high RH conditions. The solid S-BrC selected by the light yellow cycles are solid S-BrC with thin organic coatings and can uptake water. The scale bar is 1 μm . (b) Lensing enhancement of solid S-BrC cores (diameters from 100 to 800 nm, $\text{RI}_{\text{solid S-BrC},550} = 1.49 + 0.056i$) coated with water (0 – 2500 nm thickness, $\text{RI}_{\text{water},550} = 1.33 + 0i$) at 550 nm ($E_{\text{abs,CL}}$), which can vary between 1.004 and 2.851 (see Section S3). (c) O:C elemental ratio from this study and literature.^{4,8,10,16,20,38,53–55 1,5–12} We also reference this study's O:C ratio using 21-T FTICR MS data. The red lines indicate the means, the black dots are the medians, the gray rectangles are the interquartile ranges, the gray vertical lines are the 95 % confidence intervals, and the violin-shaded areas show the data distribution.”

7) Finally, the title, why there is no inorganic. Inorganic in aged tarball also contribute the optical absorption. Please consider how to revise it and widely accepted by the readers. I think that current one might make mistake your work.

We agree with the reviewer that inorganics in aged solid S-BrC can potentially contribute to light absorption. However, in response to your earlier comment 5, we elaborated that for this specific study, we observed a minor contribution of inorganics and that they might not play a major role in the water uptake. This is also surprising to us and emphasizes that our finding is novel since organics facilitate water uptake. Therefore, we decided to keep the same title.

Reviewer #2 (Remarks to the Author):

In this paper, Cheng et al. have made several high quality measurements on solid-state strongly absorptive brown carbon (solid S-BrC) in terms of composition, optical properties, and hygroscopicity. Their measurements provide variable knowledge of S-BrC, which is an important factor of climate. Thus, I have no doubt that this work will contribute to atmospheric and climate science. On the other hand, as the journal aims to "represent important advances of significance to specialists within each field", I question whether the results of this paper reach that level. They found a significant number of S-BrC from a small set of samples (two-day sampling) and showed their compositions and optical properties. These results agree with previous studies and strongly supports the previous observations using a state-of-the-art instrument. On the other hand, the results are not highly novel. In addition, although the results and discussions of the coating on S-BrC on the radiative forcing are novel, the discussions are not directly supported by their observations. Therefore, I cannot strongly support this paper being published in this multidisciplinary high impact journal.

We appreciate the reviewer for providing insightful comments and noting our high quality and valuable measurements. We understand that maybe the novelty part was not evident in the manuscript. We revised some parts of the manuscript to clarify the study's novelty. We also outline below the novel aspect of the manuscript:

- 1. Solid S-BrC with different surface layers may dictate the hygroscopicity. We showed that Solid S-BrC with an oxygenated layer can uptake water (hydrophilic Solid S-BrC), and those without are hydrophobic. This finding along with quantifying number fraction of these particles is important for parameterizing climate models' hydrophobic and hydrophilic solid S-BrC.**
- 2. Water coating on hydrophilic solid S-BrC can enhance the light absorption of solid S-BrC by a factor of two. This novel finding also suggests the current models might underestimate the climate effects of wildfire smoke.**
- 3. Solid S-BrC can be coated with organics, which enhances light absorption more than water coating.**

We understand the reviewer's concerns about the number of samples since we focused our results on limited samples collected from September 2017. However, we conducted high-resolution mass spectrometry and single particle analysis (STXM/NAXFS and CCSEM/EDX) on solid S-BrC samples collected in 2018 (see Ijaz et al.,⁵ and our response to specific comments below), and their chemical composition were similar. Therefore, we expect these samples to have solid S-BrC with similar properties, supporting our findings and having broad implications. We agree with the reviewer that our paper still has room to improve. We believe addressing these comments can help improve our manuscript significantly. Therefore, we modified the highlights of the manuscript (please see below). Below are our responses to each comment, as the reviewer suggested:

Specific

comments.

Line 69: "collected during the Pacific Northwest wildfire events on September 5 and 6, 2017" and line 403 "from September 5–6, 2017, and August 9–14, 2018,": It is unclear whether the samples were collected only in 2017 or in both 2017 and 2018. In either way, the sampling periods are limited and it is difficult to draw a general conclusion from the limited samples.

We apologize for the confusion. We have revised the sentence below to make it clear:

RL455-460: “Wildfire aerosols were collected daily from 9 AM to 2 PM with 50% duty cycle (30 mins on and 30 mins off) from September 5–6, 2017, and August 9–14, 2018, local time at the Atmospheric Measurement Laboratory in Richland, WA (46.340844 N, 119.278110 W). In this study, all analysis focuses on samples collected from September 5–6, 2017, and details about samples collected from August 9–14, 2018, have been discussed in Ijaz et al.⁴⁶”

We understand the reviewer’s concern about the sample number. Although our study focuses on the sample collected on September 5th and 6th, 2017, our findings are still novel and impactful. We collected solid S-BrC samples at different times in August 2018.⁶ The STXM/NEXAFS spectrum of solid S-BrC collected in that study is similar to that of solid S-BrC in our study, suggesting they have similar chemical composition. Therefore, these solid S-BrC could have similar hygroscopicity and optical properties. However, we acknowledge the caveat of limited samples and propose that more study is necessary to better understand our findings. To make this clear, we add the discussion below:

RL190-193: “We utilized this period because of an existing model domain setup. Moreover, that period had solid S-BrC-rich wildfire smoke events where solid S-BrC’s carbon K-edge spectrum (Fig. S5) and molecular formula similar to those in the 2017 study period,^{30,46} suggesting they have similar properties.”

Figure S5. Averaged STXM/NEXAFS spectra of individual solid S-BrC particles for 2017 solid S-BrC and 2018 solid S-BrC. The shaded area in (a-c) represents measurement uncertainties

Line 79 “We manually identified particle types based on their morphology” Please define the morphological criteria. It will also affect the discussion of the hygroscopicity study in Fig. 4.

We agree with the reviewer and have added the following sentence to the revised manuscript. This information is also included in the Fig. 1 caption in the original manuscript.

RL84-87: “We manually identified and counted the number fraction of solid S-BrC (spherical shape) and other OA particles that are not solid S-BrC (other OA, dome-like or flat shape), BC particles (fractal or compressed small monomer aggregates), and inorganic particles (crystal or irregular shape).^{20,26}”

Line 130-133: “As shown in Figs. 2a and 2b, our n_{550} and k_{550} are about 10% and 70% lower than those described in Alexander et al., respectively. This discrepancy might be due to our advanced STEM's lower electron acceleration (80 kV vs. 120 kV), which results in lower Cherenkov radiation effects and electron beam-induced knock-on damage, which can overestimate the k ”. This discussion suggests that the detailed comparison of refractive indices from this study and those of Alexander et al in Fig. 3 is meaningless since the difference is due to the uncertainties of the measurements.

We agree with the reviewer that the current discussion can be improved. However, comparing our results with those of Alexander et al. is still essential since Alexander et al.¹⁰ is the first study to use the EELS/STEM technique to estimate the optical properties of solid S-BrC. The analytical technique has improved significantly in the last decade, so the associated measurement uncertainty is reduced. Moreover, several studies still use results from Alexander et al. For example, Park et al. used the mass absorption cross section value from Alexander et al. in their chemical transport models.¹¹ Feng et al. used Alexander et al.'s RI as the RI of coating on soot to simulate the light absorption enhancement due to coating.¹² Saleh 2020 used the results of Alexander et al. to develop the BrC class based on their optical properties.¹³ We used values from Alexander et al. as the upper limit for the sensitive study (Fig. 3), which shows the importance of accurately representing the solid S-BrC optical properties to better estimate wildfire aerosol climate effects. To make that clear, we revised the sentence as below:

RL136-142: “As shown in Figs. 2a and 2b, our n_{550} and k_{550} are about 10% and 70% lower than those described in Alexander et al.¹⁸, respectively. This discrepancy might be due to our advanced STEM's lower electron acceleration voltage (80 kV vs. 120 kV). This lower electron acceleration voltage results in lower Cherenkov radiation effects and electron beam-induced knock-on damage, which can overestimate the k .^{18,34,35} Therefore, our optical properties of solid S-BrC can improve the uncertainties of solid S-BrC optical properties in the literature due to measurement limitations.”

Line 127-160: In the second paragraph of this section (line 127-139), the authors explain the difference in refractive indices due to different atmospheric processes. In the third and fourth paragraphs, they discuss the differences based on SP2 fractions and molecular compositions, respectively. Although all these processes may contribute to the refractive indices, a comprehensive discussion is helpful.

Our understanding of the facts affecting solid S-BrC's strong light absorption properties is limited. The difference in the refractive index (RI) might be due to variable sp2 fraction and molecular composition. We have included the available studies showing the sp2 fraction and molecular composition of solid S-BrC. There were no attempts to link both individual particle sp2 fraction and molecular composition with solid S-BrC RI. In our study, we comprehensively analyzed single solid S-BrC RI, sp2 fraction, and bulk sample molecular composition to investigate the association between light absorption properties of solid S-BrC and sp2 and organonitrate fraction. Although we don't have individual particle molecular composition data related to RI, this motivates future study. Our discussion provides directions

for future studies to better understand the facts that affect solid S-BrC's strong light-absorption properties. To make this clear, we add the following discussions.

RL170-174: “To conclude, our study comprehensively analyzed single solid S-BrC RI and chemical composition to investigate the association between light absorption properties of solid S-BrC, sp² fraction per particle, and the fraction of organonitrate compounds. Future studies should focus on molecular composition data related to RI from identical individual particles to better understand the facts that affect solid S-BrC's strong light-absorption properties.”

Line 174-177: “We show the percentage difference in the AAOD map from August 14 at 19:00 to 23:00 UTC, 2018. We utilized this period because of an existing model domain setup and because that period had solid S-BrC-rich wildfire smoke events with similar molecular compositions to the 2017 study period.” The authors explain why they chose a different time period between their observations and modeling; however, it is better to use the same time period to draw a solid conclusion.

We agree with the reviewer that doing the WRF-Chem simulation over the same period as the sample discussed in this manuscript would be better. However, as we mentioned in the manuscript and response to your first specific comment, the August 2018 samples have similar physical, chemical, and optical properties to the September 2017 samples. We have included additional evidence to justify this in the SI (see our response to the previous comment). Another practical issue is that we did not have the bandwidth to develop the model domain during the September 2017 sampling period, so we used the existing model domain setup. Since our main conclusion is that we need to better parameterize the solid S-BrC in the model, using different model domains might not weaken this conclusion.

Line 233-237: “We might underestimate the fraction of organic-coated solid S-BrC based on microscopy imaging because these coatings might be evaporated in the high vacuum chamber. Considering this caveat, our finding suggests that solid S-BrC can act as a seed for secondary organic aerosol formation and that the organic coating can potentially increase the contribution of solid S-BrC to climate warming.” The discussion in the second sentence is based on a hypothesis and lacks evidence. In the abstract, they mention that “Furthermore, the organic coating on solid S-BrC can lead to even higher lensing enhancements than water,” which is not supported by the evidence.

We understand the reviewer's valid concern. Indeed, we cannot observe the process of losing volatile and semi-volatile species in the SEM chamber since the electron beam does not turn on until the chamber reaches the vacuum level. However, based on the evaporation kinetic measurements on the same sample, we observed 10% volume loss after 24 hours of evaporation in an organic-free evaporation chamber.⁷ Therefore, we hypothesize that this volume loss might be partially due to the evaporation of organic coatings. We agree with the reviewer that we do not have evidence to show that the organic layer is light-absorbing. However, there are multiple studies have shown that secondary organic aerosols (SOA) can be light-absorbing^{14,15}. Although we do not have direct evidence, we utilized literature-reported value to simulate the potential lensing enhancement due to the light-absorbing organic coating. To make these points clear, we revised the sentence as below:

RL268-277 “Besides the water shell, we observed organic coating on the solid S-BrC (Fig. S10). We might underestimate the fraction of organic-coated solid S-BrC based on microscopy imaging because these coatings might be evaporated in the high vacuum chamber. Considering this caveat, our finding suggests that solid S-BrC can act as a seed

for secondary organic. Moreover, multiple studies have shown that secondary organic aerosols (SOA) can be light-absorbing.^{59,60} Thus, these light-absorbing SOA can be coated on the solid S-BrC and cause even higher lensing enhancement than water.²⁵ We estimated the lensing enhancement loss (the difference between enhancement with a clear coating and with a light-absorbing coating)²⁵ and the value to be up to a factor of 1.3 (Fig. S11). These results confirm that the organic coating can potentially increase the contribution of solid S-BrC to climate warming.⁶¹ Further study is needed to better quantify the fraction of organic coated solid S-BrC and the light-absorption enhancement due to light-absorbing SOA coating.”

Fig. 4a and S5. “solid S-BrC that uptake water” Some of them look like they are deliquescing rather than absorbing water. This result may suggest that inorganic salt that attached to or internally mixed with the S-BrC has deliquesced. If the authors have considered this possibility, I suggest adding some discussion. At least, I found many sulfate in Fig. S1.

We appreciate the reviewer’s constructive comments. We agree with the reviewer that atmospheric-aged solid S-BrC can be internally mixed with hygroscopic species such as K and sulfate (see ref¹⁻⁴). We also agree that our SEM images might not have high enough resolution to see the internal detail structure of individual particles. However, we observed that most tar balls did not have inorganics based on SEM/EDX, STXM, and TEM results. Very few particles have some inorganic inclusions, but due to the hydrophobic nature of solid S-BrC, water might not be able to diffuse into the particle. We acknowledge the potential effects, but this might not be the case in our study since our CCSEM-EDX and STXM/NEXAFS only show a negligible amount of inorganic elemental percentage ($1.3\pm 1.3\%$) and inorganic volume fraction (0.04 ± 0.09) in individual particles, respectively. Figure 4 a shows water (lighter gray, indicated by white arrow) coated on the Solid S-BrC (black, indicated by dark gray arrow) in 97%RH and 100%RH. This suggests that the Solid S-BrC does not dissolve in water after water uptake, and the water uptake happens on the surface of Solid S-BrC. However, we have not eliminated the trace amount of inorganics. We added some discussion below.

RL225-237: “It should be noted that hygroscopic solid S-BrC (indicated by white cycles) is still visible at RH>97%, suggesting that most of the volume of solid S-BrC is non-hydrophobic. Moreover, this water forms a shell on hygroscopic solid S-BrC. Although these particles did not form core-shell morphology when collected on the TEM grids, we still cannot eliminate the potential for water coatings to fully cover the airborne hygroscopic solid S-BrC due to water surface tension. Moreover, after dehydration, we found that the hygroscopic solid S-BrC did not have obvious deformation. There was a residual surrounding the solid S-BrC (Fig. 4a). Our CCSEM-EDX (Fig. S7), the Scanning Transmission Electron Microscopy Elemental map (Fig. S8) and STXM/NEXAFS Carbon speciation maps of solid S-BrC particles (Fig. S9) only show a negligible amount of inorganic elemental percentage ($1.3\pm 1.3\%$) and inorganic volume fraction (0.04 ± 0.09) in individual particles, respectively. Thus, we hypothesize that those residuals may have resulted from a thin, hygroscopic organic layer on the surface of the hydrophilic solid S-BrC particles that dissolved in water and then remained on the substrate surface after water evaporation. This hydrophilic organic layer may have resulted from condensation and deposition of polar organics and surface oxidation.”

Figure S7. Elemental percentage of individual particles analyzed by CCSEM-EDX.

Figure S8. STXM/NEXAFS Carbon speciation maps of solid S-BrC particles.

Figure S9. Elemental map of solid S-BrC acquired by Scanning Transmission Electron Microscopy (Thermo Fisher, model Titan 80-300) (STEM)/EDX with energy-dispersive X-ray spectroscopy (Oxford Instruments).

Fig. S4. Is the first question (Yes/No for $\text{Na} > 1\%$) correct?

We thank the reviewer for pointing this out. We have corrected this figure as below.

Reviewer #3 (Remarks to the Author):

Review of "Enhance light absorption from solid-state brown carbon from wildfires due to organic and water coatings" by Cheng et al.

The study focused on a wildfire event on September 5 and 6, 2017 at a single location. The researchers conducted an extensive and comprehensive analysis of the wildfire smoke, offering a very detailed picture of the physical properties of particles from this wildfire event, including optical properties and some aspects of hygroscopicity. Most of the data and analysis are high quality and should be published. However, I wonder if the paper fits within Nature Communications' mandate. This aspect is up to the Editor.

The paper consists of four main sections: 1) prevalence of solid S-BrC in wildfire smoke aerosol, 2) optical properties and chemical composition of solid S-BrC particles, 3) sensitivity study of solid S-BrC absorption aerosol optical depth using model simulations, and 4) absorption enhancement due to water-solid S-BrC interactions and organic coatings. Sections 1-3 are not novel but still significant. Section 4 presents the most novel results, but the conclusions is less convincing than the other sections.

We appreciate the reviewer noting our comprehensive, high-quality, valuable measurements and supporting the publication. We appreciated the reviewer's constructive feedback, which helped us improve our manuscript. As the reviewer suggested, we modified the highlights of the manuscript (please see below). Below are our responses to each comment:

Detailed comments:

1) Clarity and context: I had to read the paper a couple of times to appreciate what is novel in the paper. Also, the presentation feels a little disjointed in places. For example, Sections 1-3 seem separate from Section 4. I think Sections 1-3 could be a separate paper.

We understand the reviewer's concern. However, sections 1-3 are still important and related to the Section 4. Sections 1-3 highlight the importance of accurately representing solid S-BrC number concentration and optical properties, which is essential for accurately predicting the climate effects of wildfire aerosols. We have revised our Environmental Implication section as the reviewer suggested below. Moreover, Sections 1-3 also underscore the importance of including water-solid S-BrC interactions and organic coating in climate models since these sections show that there can be about 3692 ± 952 hydrophilic solid S-BrC cm^{-3} and can have higher concentration in wildfire smoke. We have revised the manuscript below to highlight the importance of sections 1-3:

RL215-224: "However, the rest of the solid S-BrC uptake water at RH above 97% (indicated by light yellow arrows, hereafter named hygroscopic solid S-BrC), suggesting a hygroscopic behavior and might potentially serve as cloud condensation nuclei (CCN) under supersaturated condition. Based on the aerosol size distribution³⁰ and MOUDI impactor collection efficiency,⁴⁸ and assuming 50% of solid S-BrC particles are hygroscopic and will also activate into droplets at supersaturation conditions, we estimated that these hygroscopic solid S-BrC could result in CCN number concentrations $3692.2 \pm 952.1 \text{ cm}^{-3}$ between 131 and 445.1 nm particle size range during the sampling period. These estimates are comparable with modeled CCN in southeast Atlantic biomass-burning aerosol-dominated region.⁴⁹ Future studies are needed to better understand the contribution of hygroscopic solid S-BrC to CCN."

RL292-305: “Our study suggests solid S-BrC can be a major component in some wildfire smoke. Without accurately representing it, climate models might underestimate the warming effect of wildfire smoke. Moreover, our finding suggests that ~50% of solid S-BrC are hygroscopic and can act as CCN at high RH environments, leading to cloud-heating effects.⁶² Thus, considering hydroscopic solid S-BrC in models might improve the predicted aerosol indirect climate effects. Previous studies primarily focus on the lensing enhancement of light-absorbing and Directive Radiative Forcing (DRF) of soot.^{63,64} Our study established a previously unrecognized concept by showing that water coating on solid BrC can cause lensing enhancement up to 2.3 at k_{550} . Since the WRF-Chem model does not include parameterization of lensing enhancement of solid S-BrC, we used the theoretical calculation to estimate the top-of-the-atmosphere DRF.⁶⁵ It shows that a 200-nm thick clear coating can lead to ~43% enhancement in 200 nm diameter solid S-BrC, enhancing directive radiative forcing at 550 nm ($E_{\text{DRF},550}$) (see section S4). Moreover, a light-absorbing organic coating can increase the lensing enhancement, leading to ~240% $E_{\text{DRF},550}$ of 200 nm diameter solid S-BrC with 200 nm thick coating. These findings should be parameterized in climate models to reduce the discrepancy between measurements and improve the model's accuracy.”

2) Title: The title only highlights Section 4. I.e., the title does not represent all the content of the manuscript.

We understand the reviewer’s valid concern but do not fully agree with it. The title should reflect the main message of the manuscript within the word limit (15 words). Still, results from other sections were also significant in making the point about the characteristics of solid S-BrC. Therefore, we decided to keep the same title.

3) Environmental implications: The environmental implications only cover the results in Section 4. This is one reason for arguing that Sections 1-3 could be a separate paper.

We thank the reviewer for providing such constructive comments. We have revised the Environmental Implications section to reflect the implications from other sections.

RL292-305: “Our study suggests solid S-BrC can be a major component in some wildfire smoke. Without accurately representing it, climate models might underestimate the warming effect of wildfire smoke. Moreover, our finding suggests that ~50% of solid S-BrC are hygroscopic and can act as CCN at high RH environments, leading to cloud-heating effects.⁶² Thus, considering hydroscopic solid S-BrC in models might improve the predicted aerosol indirect climate effects. Previous studies primarily focus on the lensing enhancement of light-absorbing and Directive Radiative Forcing (DRF) of soot.^{63,64} Our study established a previously unrecognized concept by showing that water coating on solid BrC can cause lensing enhancement up to 2.3 at k_{550} . Since the WRF-Chem model does not include parameterization of lensing enhancement of solid S-BrC, we used the theoretical calculation to estimate the top-of-the-atmosphere DRF.⁶⁵ It shows that a 200-nm thick clear coating can lead to ~43% enhancement in 200 nm diameter solid S-BrC, enhancing directive radiative forcing at 550 nm ($E_{\text{DRF},550}$) (see section S4). Moreover, a light-absorbing organic coating can increase the lensing enhancement, leading to ~240% $E_{\text{DRF},550}$ of 200 nm diameter solid S-BrC with 200 nm thick coating. These findings should be parameterized in climate models to reduce the discrepancy between measurements and improve the model's accuracy.”

4) Significance: the most novel aspect of the paper is the observation that approximately half of the solid S-BrC are hydrophobic, and half take up water at 97 % RH. Even this observation is not completely shocking since a) one previous paper by the authors observed two types of solid S-BrC in a previous field campaign, b) hydrophobic and hydrophilic solid S-BrC have been observed previously in separate studies, as pointed out by the authors, and c) a recent study observed solid S-BrC in cloud residuals (Adachi et al. Egusph. 2024) indicating that at least some solid S-BrC are hydrophilic. These previous studies may compromise the manuscript's significance.

We thank the reviewer for providing this constructive comment. Although the reviewer referred to a few great previous studies to argue that our manuscript is not significant, we believe our manuscript is novel and can add great value to the community. Please see our response below for each of the points:

a) one previous paper by the authors observed two types of solid S-BrC in a previous field campaign

China et al. 2013 observed two types of solid S-BrC, which have different electronic brightness and darkness. However, China et al. 2013 did not show the connection between electronic brightness and darkness and hygroscopicity, which is addressed by our study. To make this clear, we add the following discussion:

RL249-250: “Therefore, these studies provide evidence of two types of solid S-BrC. However, the link between different types of solid S-BrC and their hygroscopicity is still unrevealed.”

b) hydrophobic and hydrophilic solid S-BrC have been observed previously in separate studies, as pointed out by the authors

This is true, but there is an active debate about the hygroscopicity of solid S-BrC. These studies either show that solid S-BrC are hydrophilic or hydrophobic but did not report the existence of solid S-BrC with different hygroscopicity in the same aerosol population. Moreover, these studies did not provide a method to quantify the fraction of hygroscopicity solid S-BrC. Our study has shown that these two types of solid S-BrC can exist in the same aerosol population, and we might be able to identify and quantify them based on different O:C ratios.

c) a recent study observed solid S-BrC in cloud residuals (Adachi et al. Egusph. 2024) indicating that at least some solid S-BrC are hydrophilic

We thank the reviewer for pointing out this great study, which we also cited in the manuscript. Adachi et al. is the first study to report solid S-BrC in the cloud. That study shows that solid S-BrC can be active as cloud condensation nuclei. Unlike Adachi et al., our study provides laboratory observation to show that solid S-BrC has different hygroscopicity. Moreover, our study mainly focuses on the solid S-BrC's optical properties, which is not discussed in Adachi et al. Therefore, we believe this might not compromise our manuscript's significance. To improve the discussion, we modified the sentence in the introduction:

RL65-70: “Adachi et al. reported abundant solid S-BrC collected in the pyrocumulonimbus cloud to have thin layers or coatings as residual of water mixed with water-soluble species,²³ suggesting solid S-BrC could be hygroscopic and potential cloud condensation nuclei. Those coatings on hygroscopic solid S-BrC in the cloud can enhance light absorption by solid S-BrC via lensing effects. The lensing enhancement is typically solely

considered for soot^{24,25} and ignored for solid S-BrC, which might contribute to the discrepancy between models and observations.”

5) Validity of the data: I have questions about the validity of the data interpretations and conclusions, especially the results in Section 4. See below.

We appreciate the reviewer for providing these valid comments. Please see our detailed response to each point below:

6) Page 7, lines 198 to 201: "As shown in Fig. 4a, ~50% of the solid S-BrC did not uptake water (indicated by black arrows) even at >99% RH (additional images in Fig. S5). However, the rest of the solid S-BrC uptake water at RH above 97% to form a core-shell morphology with an aqueous shell (indicated by light yellow arrows, hereafter named hygroscopic solid S-BrC), suggesting a potential function as cloud condensation nuclei." Based on Fig. 4a, it doesn't look like the particles form core-shell morphology, contrary to the suggestions of the authors. It looks more like the solid S-BrC often remains at the interface of the water droplets. Also, one of the particles with a light yellow circle doesn't appear to take up any water (the particle with a light yellow circle at the top right-hand corner of the figure).

We agree with the reviewer. We have revised the sentence to not use core-shell morphology. Instead, we say water can form a shell. Although these particles did not form core-shell morphology when collected on the TEM grids, we still cannot eliminate the potential that water coatings can fully cover the airborne hygroscopic solid S-BrC due to water surface tension. We revised the paragraph as below:

RL268-277: “Besides the direct climate effects, there is limited knowledge about solid S-BrC indirect effects. Here, we conducted water-uptake experiments in an environmental SEM at 5°C and analyzed more than 200 solid S-BrC particles. As shown in Fig. 4a, ~50% of the solid S-BrC did not uptake water (indicated by black arrows) even at >99% RH (additional images in Fig. S5). However, the rest of the solid S-BrC uptake water at RH above 97% (indicated by light yellow arrows, hereafter named hygroscopic solid S-BrC), suggesting a hygroscopic behavior and might potentially serve as cloud condensation nuclei (CCN) under supersaturated condition. Based on the aerosol size distribution³⁰ and MOUDI impactor collection efficiency,⁴⁸ and assuming 50% of solid S-BrC particles are hygroscopic and will also activate into droplets at supersaturation conditions, we estimated that these hygroscopic solid S-BrC could result in CCN number concentrations $3692.2 \pm 952.1 \text{ cm}^{-3}$ between 131 and 445.1 nm particle size range during the sampling period. These estimates are comparable with modeled CCN in southeast Atlantic biomass-burning aerosol-dominated region.⁴⁹ Future studies are needed to better understand the contribution of hygroscopic solid S-BrC to CCN.”

We agree that the particle the reviewer pointed out did not uptake a significant amount of water as other hydrophilic solid S-BrC. To correct that, we removed that yellow cycle.

7) Page 7, lines 202-203: "Some hygroscopic solid S-BrC particles have an organic coating (indicated by light yellow circles), implying that hygroscopic organic coating can promote the water uptake of solid S-BrC." What is the proof of an organic coating? Is the proof the observation of a larger "residual" after evaporation? One of these particles with a light yellow circle did not leave a larger "residual" after evaporation. Also, in some cases, the residual may have been caused by a water-soluble particle nearby. Also, how can the authors differentiate between partial dissolution of the solid S-BrC and the presence of an organic coating? The solid S-BrC particles could partially or completely dissolve if enough water is added.

We apologize for not making this clear. Evidence of organic coating is found in the coating on the solid S-BrC before water uptake experiments, so we know the coating is not residual. We add one representative TEM image in the SI to show the difference between coated and uncoated solid S-BrC. Therefore, the hygroscopic solid S-BrC is a unit of hygroscopic organic coating and non-hydrophobic solid S-BrC core. Regarding the concern that solid S-BrC might dissolve in water, we believe the solid S-BrC core of hygroscopic solid S-BrC is insoluble in water. Still, their hygroscopic coating will dissolve in water. One reason is that those images at 100%RH were taken after exposure to 100%RH for more than 2 hours. After dehydration, we observed negligible change in particle size, confirming they did not dissolve in water. This agrees with Adachi et al. since they still can observe solid S-BrC in cloud droplets. We deleted the yellow circle for one of these particles with a light-yellow circle that did not leave a larger "residual" after evaporation. We agree with the reviewer that some residual after dehydration close to solid S-BrC might be caused by water-soluble particles nearby. We closely monitor the water uptake process and only count solid S-BrC particles that uptake water by themselves, not substrate, as hygroscopic solid S-BrC.

To make this clear, we add the discussion below:

RL225-237: “It should be noted that hygroscopic solid S-BrC (indicated by white circles) is still visible at RH>97%, suggesting that most of the volume of solid S-BrC is non-hydrophobic. Moreover, this water forms a shell on hygroscopic solid S-BrC. Although these particles did not form core-shell morphology when collected on the TEM grids, we still cannot eliminate the potential for water coatings to fully cover the airborne hygroscopic solid S-BrC due to water surface tension. Moreover, after dehydration, we found that the hygroscopic solid S-BrC did not have obvious deformation. There was a residual surrounding the solid S-BrC (Fig. 4a). Our CCSEM-EDX (Fig. S6), the Scanning Transmission Electron Microscopy Elemental map (Fig. S7) and STXM/NEXAFS Carbon speciation maps of solid S-BrC particles (Fig. S8) only show a negligible amount of inorganic elemental percentage ($1.3\pm 1.3\%$) and inorganic volume fraction (0.04 ± 0.09) in individual particles, respectively. Thus, we hypothesize that those residuals may have resulted from a thin, hygroscopic organic layer on the surface of the hydrophilic solid S-BrC particles that dissolved in water and then remained on the substrate surface after water evaporation. This hydrophilic organic layer may have resulted from condensation and deposition of polar organics and surface oxidation.”

Figure S10. Representative TEM image of solid S-BrC. The solid S-BrC selected by the light-yellow rectangle are solid S-BrC with thin organic coatings and can uptake water. The scale bar is 500 nm.

8) Page 8, lines 235-237: "Considering this caveat, our finding suggests that solid S-BrC can act as a seed for secondary organic aerosol formation and that the organic coating can potentially increase the contribution of solid S-BrC to climate warming." I don't see conclusive evidence for an organic coating since alternative explanations are also possible for the observations. So, I don't think the results provide convincing evidence that solid S-BrC can act as a seed for secondary organic aerosol formation.

As we explained in previous comments, the organic coating surrounds the solid S-BrC in dry conditions. Since the RH was between 20 and 70% RH, we do not expect these to be from water coating like those in Adachi et al.

RL268-277: "Besides the water shell, we observed organic coating on the solid S-BrC (Fig. S10). We might underestimate the fraction of organic-coated solid S-BrC based on microscopy imaging because these coatings might be evaporated in the high vacuum chamber. Considering this caveat, our finding suggests that solid S-BrC can act as a seed for secondary organic. Moreover, multiple studies have shown that secondary organic aerosols (SOA) can be light-absorbing.^{59,60} Thus, these light-absorbing SOA can be coated on the solid S-BrC and cause even higher lensing enhancement than water.²⁵ We estimated the lensing enhancement loss (the difference between enhancement with a clear coating and with a light-absorbing coating)²⁵ and the value to be up to a factor of 1.3 (Fig. S11). These results confirm that the organic coating can potentially increase the contribution of solid S-BrC to climate warming.⁶¹ Further study is needed to better quantify the fraction of organic coated solid S-BrC and the light-absorption enhancement due to light-absorbing SOA coating.

9) Page 9, lines 254-256. "Our study established a previously unrecognized concept by showing that water coating on solid S-BrC can cause lensing enhancement up to 2.3 at k550." This assumes the solid S-BrC is in the center of the water droplets (i.e., core-shell), and the solid S-BrC doesn't dissolve in water. However, the images (Figure 4) suggest they will not form a core-shell structure, and some of the images indicate that the solid S-BrC may have partially dissolved. In the current experiments, the water uptake is less than what will happen in a cloud droplet. Perhaps in a cloud droplet, the solid S-BrC will completely dissolve.

We thank the reviewer for providing these valid concerns. We believe the solid S-BrC core of hygroscopic solid S-BrC is insoluble in water, but their hygroscopic coating will dissolve in water. One reason is that those images at 100%RH were taken after exposure to 100%RH for more than 2 hours. After dehydration, we observed negligible change in particle size, confirming they did not dissolve in water. This agrees with Adachi et al. since they still can observe solid S-BrC in cloud droplets. As we explained, the organic coating surrounds the solid S-BrC in dry conditions. Since the ambient RH was between 20 and 70% RH during the sampling period, we do not expect these coatings to be from water like those in Adachi et al.

Figure S12. Ambient temperature and relative humidity during the sampling period. Data were reported by the meteorological measurements at the Atmospheric Measurement Laboratory of Pacific Northwest National Laboratory.

10) The authors argue that some of the solid S-BrC have an organic coating. How does this affect their optical constants measurements? I guess the authors assumed the solid S-BrC did not have an organic coating when extracting the optical constants.

We only do EELS/STEM on solid S-BrC without any coatings and inclusions. We clarify that by modifying the sentence below:

RL126-128: “To understand the direct climate effects of solid S-BrC, we probed the RI of 40 single solid S-BrC particles without any coating or inclusions as a function of wavelength (300–1000 nm) using electron energy-loss spectroscopy coupled to scanning transmission electron microscopy (EELS/STEM).”

References:

1. Adachi, K. *et al.* Spherical tarball particles form through rapid chemical and physical changes of organic matter in biomass-burning smoke. *Proc. Natl. Acad. Sci. U. S. A.* **116**, 19336–19341 (2019).
2. Adachi, K. *et al.* Occurrence, abundance, and formation of atmospheric tarballs from a wide range of wildfires in the western US. *Egusph. [preprint]* 1–30 (2024) doi:doi.org/10.5194/egusphere-2024-880.
3. Mathai, S. *et al.* Optical Properties of Individual Tar Balls in the Free Troposphere. *Environ. Sci. Technol.* **57**, 16834–16842 (2023).
4. Yuan, Q. *et al.* Evidence for Large Amounts of Brown Carbonaceous Tarballs in the Himalayan Atmosphere. *Environ. Sci. Technol. Lett.* **8**, 16–23 (2021).
5. Ijaz, A. *et al.* Molecular and physical composition of tar balls in wildfire smoke: an investigation with complementary ionisation methods and 15-Tesla FT-ICR mass spectrometry. *Environ. Sci. Atmos.* **3**, 1552–1562 (2023).
6. Ijaz, A., Kew, W., China, S., Schum, S. K. & Mazzoleni, L. R. Molecular Characterization of Organophosphorus Compounds in Wildfire Smoke Using 21-T Fourier Transform-Ion Cyclotron Resonance Mass Spectrometry. *Anal. Chem.* **94**, 14537–14545 (2022).
7. Brege, M. A., China, S., Schum, S., Zelenyuk, A. & Mazzoleni, L. R. Extreme Molecular Complexity Resulting in a Continuum of Carbonaceous Species in Biomass Burning Tar Balls from Wildfire Smoke. *ACS Earth Sp. Chem.* **5**, 2729–2739 (2021).
8. Alexander, D. T. L., Crozier, P. A. & Anderson, J. R. Brown carbon spheres in East Asian outflow and their optical properties. *Science (80-.)*. **321**, 833–836 (2008).
9. Park, R. J., Kim, M. J., Jeong, J. I., Youn, D. & Kim, S. A contribution of brown carbon aerosol to the aerosol light absorption and its radiative forcing in East Asia. *Atmos. Environ.* **44**, 1414–1421 (2010).
10. Feng, X. *et al.* Can light absorption of black carbon still be enhanced by mixing with absorbing materials? *Atmos. Environ.* **253**, 118358 (2021).
11. Saleh, R. From Measurements to Models: Toward Accurate Representation of Brown Carbon in Climate Calculations. *Curr Pollut. Rep* (2020) doi:doi.org/10.1007/s40726-020-00139-3.
12. Liu, J. *et al.* Optical properties and aging of light-absorbing secondary organic aerosol. *Atmos. Chem. Phys.* **16**, 12815–12827 (2016).
13. Xie, M. *et al.* Light Absorption of Secondary Organic Aerosol: Composition and Contribution of Nitroaromatic Compounds. *Environ. Sci. Technol.* **51**, 11607–11616 (2017).
14. Lack, D. A. & Cappa, C. D. Impact of brown and clear carbon on light absorption enhancement, single scatter albedo and absorption wavelength dependence of black carbon. *Atmos. Chem. Phys.* **10**, 4207–4220 (2010).
15. Saleh, R. *et al.* Contribution of brown carbon and lensing to the direct radiative effect of carbonaceous aerosols from biomass and biofuel burning emissions. *J. Geophys. Res. Atmos* **120**, 10,285–10,296 (2015).

We want to thank the reviewers for their constructive comments. Addressing those comments has further improved the quality of the manuscript. Below, we list each reviewer's comment (regular font), followed by our response (indented, **bold** font), followed by corresponding changes in the revised manuscript (*indented, blue font*). RL and RSL represent the line number in the revised main manuscript and SI, respectively.

Reviewer #1 (Remarks to the Author):

I carefully evaluate the response. I am happy on the revision. I basically go to accept this article in NC. Here I found one minor point.

We thank the reviewer for the positive feedback and recommending our manuscript for publication. Below are our responses to your comment:

L302-303, How did the author determine 240%. The authors should cite the sources.

Here I do want make challenge this number. The light absorbing organic coating might have shield effect instead of the light enhancement. How did the author find even bigger Eabs? How do you set up the optical refractive index of light absorbing organic? Please make more suitable implication here.

We appreciate the reviewer's comment. We also agree with the reviewer that the light absorbing coating can cause the shield effect to reduce the lensing enhancement but here were meant absorption by the entire particle. Lack and Cappa 2010¹ reported that a light-absorbing coating will increase the light absorption of black carbon (BC) since the coating itself can absorb light. The coating can still reflect and refract the light toward the BC core, causing a lensing enhancement. Therefore, compared with the light absorption of the BC core alone, the coated BC will absorb more light. This absorption enhancement is typically higher than the absorption enhancement due to a clear coating. The optical refractive index (RI) of light-absorbing organic is explained in SI Section S3. The value used in our calculation is also retrieved from Lack and Cappa ($RI_{\text{coating}} = RI_{\text{BrC},550} = 1.55 + 0.01i$).¹ We understand there are some limitations since we did not measure the RI of the organic coating, and the magnitude of absorption enhancement depends on the RI of the coating materials. Moreover, the top of the atmosphere direct radiative forcing (TOA-DRF) model is a simplified model and only considers one size of solid S-BrC core. To acknowledge these limitations, we revised the implication as below:

RL273-277: “Thus, these light-absorbing SOA can be coated on the solid S-BrC and cause even higher absorption enhancement than water coating for the entire particle via both lensing effect and absorption by the coating.²⁵ We estimated the absorption enhancement loss (the difference between lensing enhancement with a clear coating and with a light-absorbing coating)²⁵ and the value to be up to a factor of 1.3 (Fig. S11).”

RL306-309: “It should be noted that the different RI of coating, coating thickness, and core size can lead to large variations in the results. Thus, future studies to better understand the climate effects of coated solid S-BrC are necessary to parameterize our findings in climate models to reduce the discrepancy between measurements and improve the model's accuracy.”

Reviewer #2 (Remarks to the Author):

The authors have considered the reviewers' comments and provided appropriate responses and discussion in the revised paper. I agree with the authors' responses and have no further comments. I hope that this paper makes a significant contribution to the understanding of brown carbon from wildfires.

We want to thank the reviewer for accepting our manuscript.

Reviewer #3 (Remarks to the Author):

Second review of “enhanced light absorption for solid-state brown carbon from wildfires due to organic and water coatings”.

I thank the authors for taking my comments seriously. However, I still don't think the paper warrants publication in Nature – Communications, which publishes papers that represent “important advances of significance to specialists within the field”. The paper is a good one, but doesn't quite reach the standard of other papers from the same field that have been published in Nature – Communications.

We appreciate the reviewer for acknowledging the importance of this study to atmospheric science and appreciating for addressing the reviewers' comments. We understand there are still some places that might not be clear for the readers about the novelty and significance. As we responded to reviewer 2 in the previous rebuttal, we highlighted further about the novel aspects of the manuscript:

1. Hand et al.², Tivanski et al.³, and Ijaz et al.⁴ showed solid S-BrC (tar balls) can have oxygenated layer based on high-resolution microscopy chemical analysis. China et al.⁵ shows two different types of tar balls based on the electron darkness levels. Moreover, there is active debate about the hygroscopicity of the solid S-BrC. Semeniuk et al.⁶ and Adachi and Buseck⁷ show solid S-BrC does not uptake water at 100% RH. However, Hand et al. reported that solid S-BrC can uptake water at ~83% RH.² The recent study from Adachi et al. reported abundant water residual on solid S-BrC collected in the pyrocumulonimbus cloud.⁸ This study's novelty lies in connecting two different communities and providing a mechanistic understanding of the hygroscopicity of solid S-BrC and their dependency on the organic coating. We did SEM imaging, STXM analysis, STEM/EDX mapping, and STEM imaging to show that solid S-BrC can be coated by an oxygenated layer, which can uptake water (hydrophilic solid S-BrC), and those without are hydrophobic. This finding and quantifying number fraction of these particles is important for parameterizing climate models' hydrophobic and hydrophilic solid S-BrC.
2. As mentioned in previous comment, while as noted some studies (e.g., Semeniuk et al.,⁶ Adachi and Buseck,⁷ Hand et al.,² and Adachi et al.⁸) investigated the hygroscopicity of S-BrC, but their potential impact on the optical properties was missing. This study investigates the novel aspect of the lensing effect of water and organic coating on solid S-BrC light absorption properties, which is typically only considered for soot.^{1,9} We reported the novel finding that water can coat hydrophilic solid S-BrC and lead to a lensing enhancement that enhances the light absorption of solid S-BrC by a factor of two. This novel finding also suggests that current models might underestimate the climate effects of wildfire smoke.
3. No study has emphasized about organic coating on solid S-BrC. We showed direct evidence based on the STEM image that organics can be coated on solid S-BrC, which enhances light absorption.

Although our study still has some limitations, we believe these novelties are important for the broader atmospheric science community and general audiences since our findings can lead to future studies to improve the climate model prediction of aerosol climate effects. Moreover, our results are an

important but missing piece in the current climate models. Including our findings in the climate model can improve the uncertainties in predicted aerosol climate effects. However, we agree with the reviewer that our paper can be further improved to emphasize the novelty and significance of the work. Therefore, we modified the manuscript based on the reviewer's comments. Please see our point-to-point response below:

Specific comments:

1) The clarity of the paper still needs to be improved, especially if the authors are considering a higher impact and more general journal. In several places it is still not clear what is new and what has already been done.

We have further enhanced the clarity and novelty of the work and revised the manuscript accordingly

RL60-64: “These studies show discrepancies in solid S-BrC hygroscopicity, and the relative abundance of hygroscopic and hydrophilic solid S-BrC in the atmosphere is still missing. Moreover, it has been reported that coating on soot can enhance the light absorption properties of soot.^{24,25} However, the effects of these coatings on solid S-BrC light absorption properties have not been investigated, which might contribute to the aerosol optical properties discrepancy between models and observations.”

RL67-79: “Here, we report a comprehensive single-particle and molecular-level analysis of solid S-BrC particles collected during the Pacific Northwest wildfire events on September 5 and 6, 2017, where >90% of particles were solid S-BrC. Given this composition, this event provides a unique opportunity to probe the physical, chemical, and optical properties of solid S-BrC. The experimentally retrieved solid S-BrC optical properties and mass fractions were used as inputs to the Weather Research and Forecasting Model coupled to chemistry (WRF-Chem) to estimate their absorption aerosol optical depth (AAOD) over the Pacific Northwest region. Additionally, we investigated the interactions between solid S-BrC and estimated the lensing enhancement due to water coating. Our results show that solid S-BrC can dominate in wildfire smoke. We found that ~50% of solid S-BrC particles can uptake water above 97% RH, which results in a lensing enhancement at 550 nm by a factor of 2. Furthermore, the light-absorbing organic can coat solid S-BrC, leading to even higher absorption enhancements than water. Additionally, we compare compositional results from a wildfire-impacted plume in the Pacific Northwest (August 2018) to assess the broader applicability of our findings across the region.²⁶”

2) the number of samples still seems too small to make strong general comments.

We understand the reviewer's concern regarding the number of samples, as most of our results focus on samples collected in September 2017. Although it may not have been explicitly clear in the manuscript, we also compared our findings with samples collected in 2018 (see Ijaz et al.,⁴), and their chemical composition was consistent. Thus, we expect these samples to contain solid S-BrC with similar properties, supporting our conclusions and extending the implications of our findings. Additionally, as the reviewer acknowledged, we have conducted an extensive analysis of these samples using multiple, often labor-intensive techniques, including miniSPLAT, EELS/STEM, STXM/NAXFS, and STEM/EDX. In this study, we performed:

- i. High-resolution mass spectrometry assigned 9,841 molecular formulas for organic compounds to probe molecular properties.
- ii. Single-particle mass spectrometry analysis of individual particles (more than 30,000 particles) to probe individual particle's chemical composition.
- iii. STXM/NAXFS analysis of 196 to probe the carbon and oxygen K-edge spectrum and mixing state of OC, IN, and EC
- iv. CCSEM/EDX analysis of 3,360 solid S-BrC particles to probe individual particles' morphology and elemental composition.
- v. EELS/STEM analysis on 40 solid S-BrC particles to probe individual solid S-BrC's RI.
- vi. STEM imaging and STEM/EDX mapping on over 100 particles to probe their detailed structure and elemental mapping.

While we agree with the reviewer that additional data could strengthen and generalize the conclusions, the comprehensive analyses of both 2017 and 2018 samples provide a representative view of the Pacific Northwest wildfire emissions.

3) Significance: In most cases, the results are not new. The new aspects do not seem significant enough for a higher impact more general journal.

We have already highlighted the novelty of our results and findings in response to the previous comment. As emphasized, the discoveries from this study represent a significant advancement in the field and offer valuable guidance to the modeling community regarding the incorporation of water and organic coatings' influence on the light absorption properties of solid S-BrC.

Our systematic and comprehensive analyses reveal three key findings: (1) solid S-BrC can exhibit hygroscopic behavior, and we quantified the fraction of hydrophilic and hydrophobic particles; (2) water coatings on solid S-BrC can enhance light absorption through the lensing effect; and (3) organic coatings on solid S-BrC result in even greater light absorption enhancement than water coatings.

These novel findings, combined with our measurements of solid S-BrC concentrations and their optical properties, underscore the critical role of solid S-BrC in the climate system. It is important to note that these insights have not yet been incorporated into current climate models. Therefore, our findings offer important opportunities for future experimental and modeling studies to better understand the climate impacts of solid S-BrC.

4) Some of the implications are uncertain due to speculation. For example, the authors assume a core-shell model when discussing the effect of solid S-BrC on cloud droplets, but whether or not this is valid is not clear.

We appreciate the reviewer for pointing this out. We have revised the manuscript to acknowledge these limitations.

RL262-268: “We utilized the model developed by Bond et al.²⁴ to estimate the absorption enhancement (E_{abs}) at 550 nm wavelength of solid S-BrC with diameters of 100–800 nm and water coating thicknesses between 0 and 2.5 μm , using our measured refractive index (RI) of solid S-BrC at 550 nm (Fig. 4b). We assume the solid S-BrC core is located at the center after uptaking water since solid S-BrC found in pyrocumulonimbus cloud droplets

can located at the center of droplets.²³ We acknowledge this assumption might overlook the effects of the possibility that the core might not located at the center, which is worth future investigation. As shown in Fig. 4b, water shell can enhance the light-absorption of solid S-BrC by up to a factor of 2.3.”

RL306-309: “It should be noted that the different RI of coating, coating thickness, and core size can lead to large variations in the results. Thus, future studies to better understand the climate effects of coated solid S-BrC are necessary to parameterize our findings in climate models to reduce the discrepancy between measurements and improve the model's accuracy.”

I do think the paper is a nice contribution to the field of atmospheric science and should be published. I also commend the authors for the large body of work.

We thank the reviewer for acknowledging our contribution to atmospheric science and supporting its publication. We hope our response clarifies the study’s novelty and significance.

References

1. Lack, D. A. & Cappa, C. D. Impact of brown and clear carbon on light absorption enhancement, single scatter albedo and absorption wavelength dependence of black carbon. *Atmos. Chem. Phys.* **10**, 4207–4220 (2010).
2. Hand, J. L. *et al.* Optical, physical, and chemical properties of tar balls observed during the Yosemite Aerosol Characterization Study. *J. Geophys. Res. Atmos.* **110**, D21210 (2005).
3. Tivanski, A. V., Hopkins, R. J., Tyliczszak, T. & Gilles, M. K. Oxygenated interface on biomass burn tar balls determined by single particle scanning transmission X-ray microscopy. *J. Phys. Chem. A* **111**, 5448–5458 (2007).
4. Ijaz, A. *et al.* Molecular and physical composition of tar balls in wildfire smoke: an investigation with complementary ionisation methods and 15-Tesla FT-ICR mass spectrometry. *Environ. Sci. Atmos.* **3**, 1552–1562 (2023).
5. China, S., Mazzoleni, C., Gorkowski, K., Aiken, A. C. & Dubey, M. K. Morphology and mixing state of individual freshly emitted wildfire carbonaceous particles. *Nat. Commun.* **4**, 1–7 (2013).
6. Semeniuk, T. A., Wise, M. E., Martin, S. T., Russell, L. M. & Buseck, P. R. Hygroscopic behavior of aerosol particles from biomass fires using environmental transmission electron microscopy. *J. Atmos. Chem.* **56**, 259–273 (2007).
7. Adachi, K. & Buseck, P. R. Atmospheric tar balls from biomass burning in Mexico. *J. Geophys. Res.* **116**, 2–8 (2011).
8. Adachi, K. *et al.* Occurrence, abundance, and formation of atmospheric tarballs from a wide range of wildfires in the western US. *Atmos. Chem. Phys.* **24**, 10985–11004 (2024).
9. Bond, T. C., Habib, G. & Bergstrom, R. W. Limitations in the enhancement of visible light absorption due to mixing state. *J. Geophys. Res. Atmos.* **111**, 1–13 (2006).